# Determinants of epidemic size and the impacts of lulls in seasonal influenza virus circulation

Simon P. J. de Jong [1], Zandra C. Felix Garza[1], Joseph C. Gibson[1], Sarah van Leeuwen[1], Robert P. de Vries[2], Geert-Jan Boons[2,3,4,5], Marliek van Hoesel[1], Karen de Haan[1], Laura E. van Groeningen[1], Katina D. Hulme[1], Hugo D. G. van Willigen [1], Elke Wynberg[1,6], Godelieve J. de Bree[7], Amy Matser[6], Margreet Bakker[1], Lia van der Hoek [1], Maria Prins[6,7], Neeltje A. Kootstra [8], Dirk Eggink[1,9], Brooke E. Nichols [1,10], Alvin X. Han [1,11], Menno D. de Jong[1,11] & Colin A. Russell [1,10,11] ✉

During the COVID-19 pandemic, levels of seasonal influenza virus circulation were unprecedentedly low, leading to concerns that a lack of exposure to influenza viruses, combined with waning antibody titres, could result in larger and/or more severe post-pandemic seasonal influenza epidemics. However, in most countries the first post-pandemic influenza season was not unusually large and/or severe. Here, based on an analysis of historical influenza virus epidemic patterns from 2002 to 2019, we show that historic lulls in influenza virus circulation had relatively minor impacts on subsequent epidemic size and that epidemic size was more substantially impacted by season-specific effects unrelated to the magnitude of circulation in prior seasons. From measurements of antibody levels from serum samples collected each year from 2017 to 2021, we show that the rate of waning of antibody titres against influenza virus during the pandemic was smaller than assumed in predictive models. Taken together, these results partially explain why the re-emergence of seasonal influenza virus epidemics was less dramatic than anticipated and suggest that influenza virus epidemic dynamics are not currently amenable to multi-season prediction.

Seasonal influenza viruses typically cause annual epidemics worldwide, infecting up to 35% of the human population[1,2]. However, the incidence of seasonal influenza was unusually low during the first two years of the COVID-19 pandemic[3–10], likely due to non-pharmaceutical interventions (NPIs) aimed at reducing transmission and spread of SARS-CoV-2, which were also effective in limiting exposure to seasonal influenza viruses[4–12]. This global lull in influenza virus circulation and consequent lack of exposure to influenza viruses led to widespread concerns, supported by

[1]Department of Medical Microbiology & Infection Prevention, Amsterdam University Medical Centers, University of Amsterdam, Amsterdam, The Netherlands. [2]Department of Chemical Biology and Drug Discovery, Utrecht Institute for Pharmaceutical Sciences, Utrecht University, Utrecht, The Netherlands. [3]Complex Carbohydrate Research Center, University of Georgia, Athens, GA, USA. [4]Bijvoet Center for Biomolecular Research, Utrecht University, Utrecht, The Netherlands. [5]Department of Chemistry, University of Georgia, Athens, GA, USA. [6]Department of Infectious Diseases, Public Health Service of Amsterdam, Amsterdam, The Netherlands. [7]Department of Infectious Diseases, Amsterdam University Medical Centers, University of Amsterdam, Amsterdam, The Netherlands. [8]Department of Experimental Immunology, Amsterdam University Medical Centers, University of Amsterdam, Amsterdam, The Netherlands. [9]Centre for Infectious Disease Control, National Institute for Public Health and the Environment, Bilthoven, The Netherlands. [10]Department of Global Health, School of Public Health, Boston University, Boston, MA, USA. [11]These authors contributed equally: Alvin X. Han, Menno D. de Jong, Colin A. Russell. ✉e-mail: c.a.russell@amsterdamumc.nl

modelling studies, that increased susceptibility to seasonal influenza viruses due to waning immunity could result in larger and more severe epidemics in subsequent seasons[7,10,12–15].

While a comparison to pre-pandemic epidemic sizes is difficult because testing behaviour in many countries has changed due to the COVID-19 pandemic, there is little evidence to suggest that the first post-pandemic influenza season was unusually large or severe. For example, the level of influenza virus circulation in the 2022 Australia influenza season was described as moderate, with low clinical severity[16]. Preliminary estimates of the 2022/2023 influenza burden in the United States suggest that the winter epidemic was not unusually severe, falling well within the range of influenza epidemics in the US prior to the COVID-19 pandemic: in 5 of 10 seasons in the previous decade, the estimated upper bound of the number of influenza hospitalisations was higher than in the 2022/2023 season, and in 6 of 10 seasons in the previous decade the lower bound was higher[17,18]. In the United Kingdom, too, rates of influenza-like illness (ILI) and influenza-attributable mortality fell well within the range observed in the decade preceding the COVID-19 pandemic[19].

The lack of a clear post-pandemic increase in season size and/or severity suggests that our current understanding of the determinants of an epidemic's size and severity, reflected in predictive models, does not capture one or more aspects of influenza virus epidemiology. Here, we sought to explain the apparent conflict between model predictions and observed epidemiological dynamics. First, we analysed two decades of epidemiological data from 47 countries to investigate the relationship between the magnitude of incidence in prior seasons and subsequent epidemic size. Second, we analysed serum samples collected longitudinally before and during the COVID-19 pandemic from adults living in the Netherlands to investigate the extent to which measured influenza antibody waning rates agree with those assumed in predictive models. Together, our analyses provide explanations for the disparity between the predicted and observed post-pandemic epidemiological dynamics of seasonal influenza.

## Results

### The effects of past (sub)type lulls on subsequent (sub)type epidemic size

First, we investigated to what extent the predicted relationship between the magnitude of influenza virus circulation in prior years and the size and severity of subsequent epidemics holds, where little activity in prior seasons should translate to a bigger subsequent epidemic. Prior to the COVID-19 pandemic, seasonal influenza virus circulation was highly heterogeneous, with individual influenza epidemics in any given country typically being dominated by one or two influenza virus (sub)types. Hence, there were frequent lull periods lasting 1–3 years where other seasonal influenza virus (sub)types barely circulated. Due to the lack of immunological cross-reactivity between (sub)types, these lulls are potentially analogous to the scenario observed during the COVID-19 pandemic for individual (sub)types. We leveraged these historical lulls to gain insight into how influenza virus circulation lulls affected subsequent influenza epidemic sizes in the past.

To identify and estimate the frequency of (sub)type lulls, we analysed virological surveillance data for 47 countries in the Northern and Southern Hemispheres for the period from 2002 until 2019, deposited in the WHO FluNet database[20] (Supplementary Table 1). (Sub)type lull periods were identified if a particular (sub)type did not exhibit substantial levels of circulation over consecutive seasons in a given country. We defined substantial (sub)type circulation for a season if the (sub)type accounted for ≥20% of detections in the country during said season. In 45%, 45%, and 77% of country-season pairs for A/H3N2, A/H1N1pdm09, and B viruses, respectively, (sub)type lulls lasted for at least one season, with some lull periods lasting as long as two seasons (Fig. 1a). Hence, extended periods of relative absence of individual influenza (sub)types are a regular feature of influenza epidemic dynamics.

While virological surveillance data demonstrates the frequency of (sub)type lulls, it does not reveal how (sub)type lulls affect epidemic size as viral sampling rates may vary from year to year. Hence, we required a metric that more accurately represents the size of a (sub)type's epidemic in a particular season, rather than solely the proportion of a season's total activity attributable to the (sub)type. To that end, we estimated (sub)type-specific relative epidemic sizes by integrating virological surveillance data with influenza-like illness (ILI) data from the WHO FluID[21] database for 20 countries in Europe and the Middle East where, in addition to the virological surveillance data described above, high-resolution ILI data was available (Supplementary Table 1). Because high-resolution ILI data is sparse for most seasons preceding the 2009 A/H1N1pdm09 pandemic, we restricted this analysis to the seasons from 2010/2011 until 2019/2020. In our estimates, a relative size of one corresponds to the mean number of influenza virus infections in a single season for a given country, irrespective of (sub)type. Very small or absent (sub)type-specific epidemics (defined as relative epidemic sizes <0.1) were observed for 28%, 23%, and 37% of country-seasons for A/H3N2, A/H1N1pdm09 and influenza B viruses, respectively (Fig. 1b).

To investigate the effect influenza virus (sub)type lulls had on epidemic composition and size, we correlated our computed (sub)type-specific relative sizes with the number of seasons elapsed since the (sub)type's previous substantial circulation. Additionally, we correlated relative size against previous season relative size and the sum of the two previous seasons' relative size. We found that both the probability of a (sub)type's substantial circulation and the mean epidemic sizes for each influenza virus (sub)type increased with time since substantial circulation (Fig. 1c, d). Nevertheless, epidemic sizes varied substantially for each value of seasons since substantial circulation (Fig. 1d). This suggests that there is an effect of absence or presence of circulation in previous seasons on epidemic size, but also that there is substantial background variation in epidemic size, independent of absence or presence of circulation in preceding seasons. Similarly, while there is a negative relationship between the relative epidemic size of each (sub)type and its relative size in the preceding season and the relative summed size over the last two seasons, there is wide variation in epidemic size: seasons with very low and very high relative sizes both occurred frequently following seasons of low-to-mid incidence (Fig. 1e). Notably, in 9 of the 20 countries included in our dataset, the first season after the 2009 A/H1N1pdm09 pandemic that saw substantial circulation of the A/H3N2 subtype (2011/2012) was not one of the three largest A/H3N2 epidemics in the influenza seasons from 2010/2011 until 2019/2020, despite three years of near-absent circulation.

Importantly, for each number of seasons since substantial circulation, we observed a striking degree of clustering of relative epidemic sizes across countries by season, suggesting the existence of season-specific effects on epidemic size, shared among countries in a single season (Fig. 1d). For example, the A/H3N2 epidemic size in 2016/2017 appeared consistently greater than in 2013/2014 despite, in many countries, equal time since previous substantial A/H3N2 circulation. We thus hypothesised that the size of (sub)type-specific epidemics could be jointly explained by a combination of season-specific effects, shared among countries, and effects related to the presence or absence of that virus (sub)type in the seasons preceding an epidemic. We used a Bayesian hierarchical model to estimate the likely effects of (i) seasons since substantial circulation, (ii) size in the previous season and (iii) the sum of previous two seasons' sizes. (Fig. 2). We also estimated the respective season-specific effects, which correspond to the predicted 'base size' of a (sub)type's epidemic given that (i) the previous substantial circulation of that (sub)type was in the previous season, (ii) there was no circulation of that (sub)type in the previous season, or (iii) there was no circulation in the previous two seasons. These season effects are modulated by the effects of prior circulation

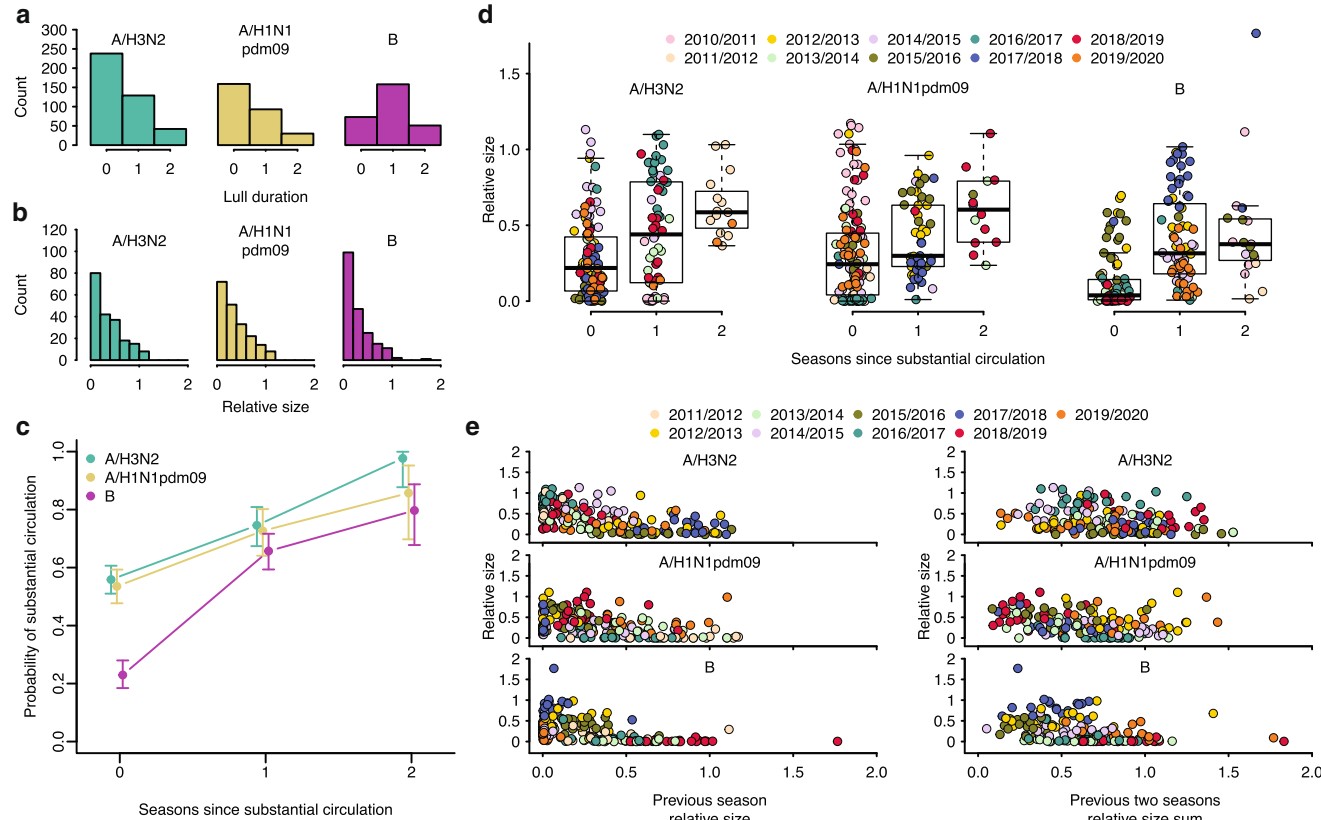

**Fig. 1 | The effects of influenza epidemic dynamics in preceding seasons on subsequent epidemic composition and size. a** The distribution of lull period durations, in number of seasons, by (sub)type, across all countries and seasons. A lull duration of zero corresponds to the same (sub)type's substantial circulation in the previous season and the current season. **b** The distribution of relative epidemic sizes by virus (sub)type, across all countries and seasons. **c** The relationship between the number of seasons since previous substantial circulation of a (sub) type and the probability of the (sub)type's substantial circulation. Error bars correspond to 95% confidence interval from an exact two-tailed binomial test for proportions. Points correspond to point estimates of the probability. The probabilities were computed from all countries and seasons (n = 642, 460 and 624 country-season pairs for A/H3N2, A/H1N1pdm09, B, respectively). **d** Relationship between the relative size of a (sub)type-specific epidemic and the number of

seasons since previous substantial circulation of that (sub)type. Each point corresponds to a specific country, in a specific season (n = 188, 198 and 186 country-season pairs for A/H3N2, A/H1N1pdm09 and B, respectively). Points are coloured by the season. Boxplots show the median and first and third quantiles. Whiskers correspond to the minimum of the maximum value and the third quartile + 1.5 × the interquartile range for the upper whisker, and the maximum of the minimum value and the first quartile − 1.5 × the interquartile range for the lower whisker.
**e** Relationship between the relative size of a (sub)type's epidemic and (1) the size of that (sub)type's epidemic in the previous season (left); and (2) the sum of the two previous seasons' sizes of that (sub)type (right). For each of the two subpanels, each point corresponds to a specific country, in a specific season (n = 180 country-season pairs for each (sub)type for previous season size, n = 160 for each (sub)type for previous two seasons size sum). Points are coloured by the season.

to yield an epidemic's predicted size. Each of the three predictors individually had non-trivial effects on epidemic size in models with season effects and estimated effects were substantially smaller than in models that did not include season effects (Fig. 2).

Crucially, models that included season effects exhibited much better predictive performance than models without season effects (Supplementary Fig. 1), showing that season effects are a crucial determinant of epidemic size. Across all model formulations, the estimated season effects, shared among countries, differed substantially between seasons. Furthermore, between-season differences in season effects were consistently substantially greater in magnitude than any of the predictors related to prior incidence. For example, in the model that includes previous season size as predictor for A/H3N2 epidemic size, the estimated season effects ('base sizes') ranged from 0.17 (95% CI 0.04–0.31) in 2015/2016 to 0.83 (95% CI 0.75–0.92) in 2016/2017: a difference of 0.66. Conversely, assuming the size of the previous season was the mean A/H3N2 season relative size (across all included countries and seasons), the effect of previous season size would only reduce predicted size by 0.06 (95% CI 0.01–0.12) compared to if there were no circulation in the previous season. Together, these results suggest that an effect of the magnitude of influenza virus

circulation in the preceding season(s) on subsequent epidemic size is present but limited and that epidemic size is dominated by season-specific factors, unrelated to the magnitude of prior circulation.

## Effects of past (sub)type lulls on subsequent influenza season severity

In addition to season size, we investigated how influenza circulation lulls affected season severity. Here, we used excess mortality as a proxy for severity, leveraging Europe-wide estimates of excess mortality as calculated by the EuroMOMO network[22,23]. By comparing these estimates to our computed lull durations, we qualitatively investigated if a clear relationship exists between prior incidence and season severity. Rates of pooled Europe-wide influenza-attribute excess mortality as calculated by the EuroMOMO network varied substantially between seasons, ranging from 0.31 (95% CI 0.24–0.38) per 100,000 in 2013/2014 to 28.58 (95% CI: 28.22–28.95) per 100,000 in 2014/2015. Hence, in the decade prior to the COVID-19 pandemic, epidemics could differ by up to two orders of magnitude in their severity[22,23]. In the 2011/2012 season, which was A/H3N2-dominant Europe-wide and followed a three-year A/H3N2 lull in almost all countries, Europe-wide total excess mortality in the winter period amounted to 6.73 (95% CI 5.26–8.21) per

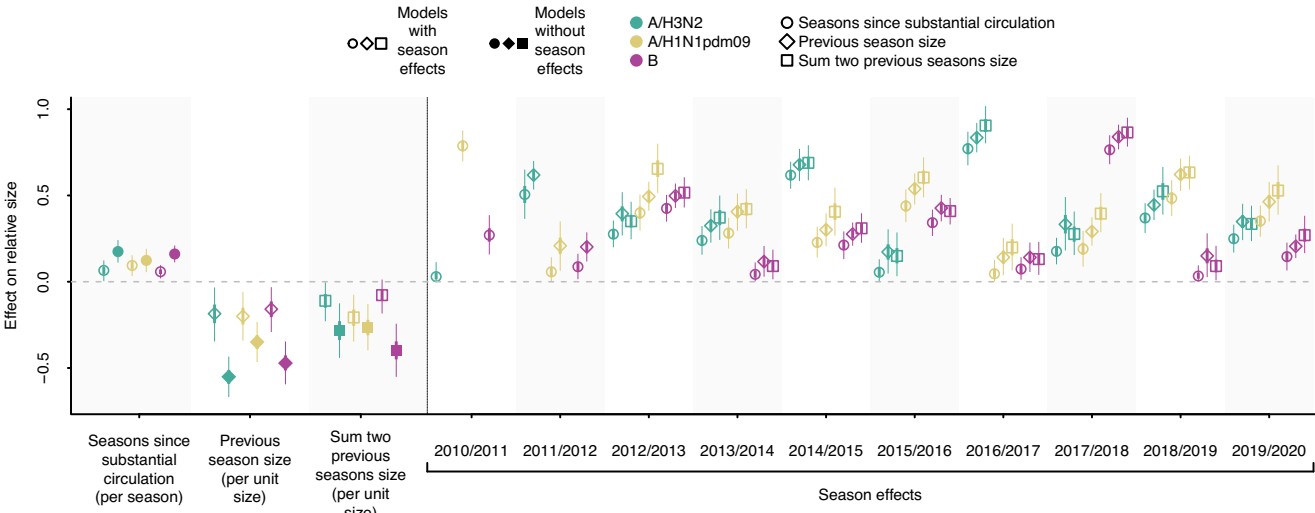

**Fig. 2 | Bayesian hierarchical model correlating influenza (sub)type relative epidemic size with seasons since substantial circulation, epidemic sizes in previous seasons and season-specific effects.** Posterior distributions of parameter estimates in the model, with seasons since previous substantial circulation (circles), previous epidemic size (diamonds), or sum of previous two epidemics' size (squares) as predictors, either with or without season effects. Points, thick and thin lines correspond to the posterior median, 50% CI, and 95% CI, respectively ($n = 188, 198, 191$ country-season pairs for A/H3N2, A/H1N1pdm09 and B, respectively, for models with seasons since substantial circulation as predictor, $n = 180$ country-season pairs for each (sub)type for models with previous season size as predictor, $n = 160$ country-season pairs for each (sub)type for models with sum of two previous seasons size as predictor).

100,000. In turn, in the 2014/2015 and 2016/2017 seasons, which were also A/H3N2-dominant Europe-wide and saw substantial A/H3N2 circulation one or two seasons prior in almost all countries, respectively, influenza-specific excess mortality amounted to 28.58 (95% CI: 28.22–28.95) and 25.65 (95% CI: 25.26–26.05) per 100,000, respectively. Hence, in these seasons, influenza-specific excess mortality was four to five-fold higher than total winter period excess mortality in 2011/2012, despite substantially shorter lull durations. While this coarse analysis could only be performed for seasons dominated by a single (sub)type, these results suggest that there is no clear relationship between the magnitude of circulation in the preceding seasons and the severity of subsequent seasons.

### Antibody responses to seasonal influenza virus during the COVID-19 pandemic

Waning of pre-existing immunity due to lack of immune stimulation has been posited to lead to larger post-lull epidemics, but evidence is lacking on precisely how antibody immunity against seasonal influenza viruses changes due to near-absence of seasonal influenza circulation, such as seen during the COVID-19 pandemic. To quantify the effects of lack of influenza virus circulation on antibody titres against seasonal influenza viruses, we analysed influenza virus antibody dynamics in the pre- and intra-COVID-19 pandemic periods in the Netherlands. We quantified the baseline antibody titres of an adult population in the Netherlands for the seasons preceding the COVID-19 pandemic and the extent of their decrease during the pandemic. Influenza A/H3N2, A/H1N1pdm09 and B/Yamagata viruses had caused epidemics in the three influenza seasons prior to the onset of the COVID-19 pandemic (Fig. 3a) and epidemic activity during this period was consistent with patterns from 2010–2019 (Supplementary Fig. 2). Since antibody responses to the haemagglutinin protein of influenza viruses are known to be correlates of protection[24–26], we measured antibody titres with haemagglutination inhibition (HI) assay against representative strains of each (sub)type of seasonal influenza in 130 serum samples from a longitudinal cohort of 31 female and 34 male adult COVID-19 patients that were not vaccinated for seasonal influenza in 2020 (the Viro-immunological, clinical and psychosocial correlates of disease severity and long-term outcomes of infection in SARS-CoV-2 – a prospective cohort study (RECoVERED))[27] (Supplementary Figs. 3 and 4a).

Additionally, we measured antibody titres against longitudinal samples collected in the summers from 2017 to 2020 from 100 healthy male adults within the Amsterdam Cohort Studies on HIV infection and AIDS[28] (ACS) (Supplementary Figs. 3 and 4b). This cohort only consists of men, and influenza vaccination status was not known, but it crucially allows for comparison of intra-pandemic against pre-pandemic influenza antibody dynamics, and hence provides important additional data. Hence, our data consisted of a total of 630 serum samples across both cohorts. Importantly, all participants were healthy, and specifically all ACS individuals were HIV-seronegative.

From 2019 to 2021, mean HI titres remained largely unchanged for all influenza virus (sub)types, including during the COVID-19 pandemic period, for both the ACS and RECoVERED cohorts (Fig. 3b and Supplementary Fig. 5b). For all seasonal influenza virus (sub)types, mean HI titres increased after the 2017/2018 influenza epidemic but returned to pre-2017/2018 levels by summer 2019 in the ACS cohort (Fig. 3b, Supplementary Fig. 5b). Due to experimental variation resulting from differences in the receptor-destroying enzyme used, the ACS individuals were split into two separate groups of 70 and 30 individuals. Results from the larger group are presented in the main text while the smaller group of 30 is referred to the Supplementary Material (Supplementary Fig. 5). Differentiating the year-on-year individual HI titre distributions by titre rises that are indicative of recent influenza virus infection ($\geq$4-fold increase, $\geq$2 $\log_2$ units), showed that influenza A and B virus infections were most common in individuals with low antibody titres in the year prior to infection (Fig. 3c and Supplementary Fig. 5c); consistent with lower antibody titres being associated with greater risk of infection. Overall, the HI titre distributions of the cohort remained largely unchanged over the study period, including during the first two years of the COVID-19 pandemic.

We applied a mathematical model on the HI titres of participants in 2020 and 2021 to estimate pandemic-period antibody titre waning rates. For the ACS individuals, we estimated that antibody titres against A/H3N2 viruses waned at $-0.06$ $\log_2$ units per year, 95% credible interval (CI) ($-0.18$, $0.05$); A/H1N1pdm09 viruses at $-0.01$, 95% CI ($-0.14$, $0.13$); B/Yamagata viruses at $0.10$, 95% CI ($-0.02$, $0.22$); and B/Victoria viruses at $0.10$, 95% CI ($-0.04$, $0.24$) (Fig. 3d, Supplementary Fig. 5d). For the RECoVERED cohort, we estimated mean waning rates towards A/H3N2, A/H1N1pdm09, B/Yamagata, and B/Victoria to be

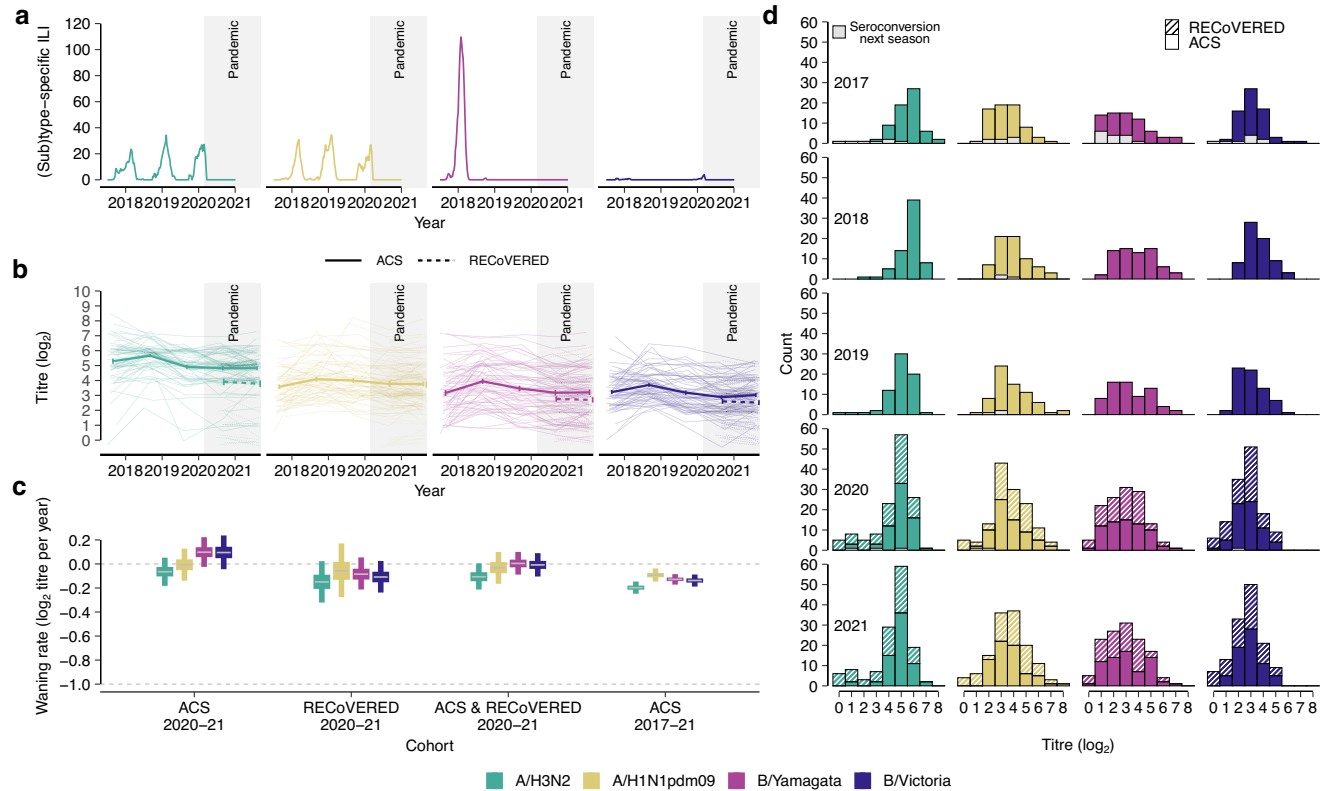

**Fig. 3 | Antibody dynamics to seasonal influenza virus before and during the COVID-19 pandemic. a** Seasonal influenza virus epidemic activity 2017–2021 in the Netherlands, computed as the number of reported cases of influenza-like illness per 100,000, stratified by (sub)type as estimated from virological surveillance data. **b** Individual antibody titres against seasonal influenza viruses based on haemagglutination inhibition (HI) assay from 2017–2021 among 70 healthy male adult participants of the Amsterdam Cohort Studies on HIV infection and AIDS (ACS) cohort for each influenza virus (sub)type as well as 34 male and 31 female participants of the RECoVERED cohort for years 2020-21 (dashed). Mean antibody titre changes across all individuals are drawn in bold lines with error bars indicating the mean standard error. The viruses used are A/Netherlands/04189/2017 (A/H3N2), A/Netherlands/10218/2018 (A/H1N1pdm09), B/Netherlands/04136/2017 (B/

Yamagata), and B/Netherlands/00302/2018 (B/Victoria). **c** HI titre distributions in the two cohorts following each winter epidemic period coloured by influenza virus (sub)type. HI titre distributions of individuals who experienced $a \geq 2$ log$_2$ units increase in HI titre (≥4-fold increase in HI titre) in the next winter epidemic period, indicating likely infection, are shown in grey bars. **d** Mean HI antibody titre waning rates by influenza virus (sub)type in adults estimated from HI titres from the individuals among the 70 ACS individuals that did not see a $\geq 2$ log$_2$ units increase in titre in consecutive years in the study period and 65 RECoVERED participants. Error bars correspond to the 50% and 95% credible intervals and horizontal grey lines correspond to the median of the posterior distribution of the mean. Waning rate of −1.0 corresponds to 1 2-fold decrease in antibody titre in 1 year.

−0.15, 95% CI (−0.32, 0.02), −0.06, 95% CI (−0.28, 0.17), −0.08, 95% CI (−0.21, 0.05) and −0.11, 95% CI (−0.24, 0.02) log$_2$ units per year respectively, in agreement with those derived from the ACS cohort (Fig. 3d). Combining data from both cohorts for the 2020–2021 period, the estimated mean waning rates remained similar to previous estimates (Fig. 3d). We also estimated mean waning rates using HI titres from the same ACS individuals for the entire 2017–2021 period (Fig. 3d, Supplementary Fig. 5d). For this period, waning estimates are generally lower with narrower credible intervals as they were estimated from longitudinal data spanning five years, but no substantial waning of HI titres against any of the viruses was observed either, and estimates were similar to estimates for the 2020–2021 period, for both the ACS and the RECoVERED cohorts (A/H3N2: −0.20, 95% CI (−0.25, −0.15), A/H1N1pdm09: −0.09, 95% CI (−0.15, −0.04), B/Yamagata: −0.13, 95% CI (−0.17, −0.08), B/Victoria: −0.14, 95% CI (−0.19, −0.09)). For the ACS cohort, we included only individuals who experienced no $\geq 2$ log$_2$ unit increases in HI titre for the entire study period and hence were likely not infected in the 2017–2021 period in our waning model (A/H3N2: $n = 59$, A/H1N1pdm09: $n = 54$, B/Yamagata: $n = 53$, B/Victoria: $n = 58$).

To investigate potential age or sex-specific patterns in antibody dynamics in our two cohorts, we stratified baseline antibody titres by age and sex, for each (sub)type. However, we found no consistent age- or sex-related effects on baseline titres (Fig. 4a, Supplementary Fig. 6a).

Similarly, we investigated if there were age or sex-specific effects on estimated individual-level waning rates, but we found no consistent age or sex-related differences (Fig. 4b, Supplementary Fig. 6b). The estimated standard deviation of the measured titre value around the model-estimated individual-level titre amounted to 0.31 log$_2$ units (95% CI 0.29–0.33) for the ACS cohort for the full 2017–2021 dataset and 0.05 (95% CI 0.03–0.08) for the RECoVERED cohort, suggesting that the model used to estimate the waning rates fits the data well.

## Discussion

Our analysis of two decades of epidemiological data from 47 countries demonstrates that low country-level prevalence of influenza (sub) types over one or more years was not unique to the COVID-19 pandemic but occurred frequently in the past. Additionally, while our analysis shows that periods or low or near-absent circulation of particular (sub)types on average led to increased epidemic sizes of that (sub)type, both very large and very small epidemics occurred following sustained periods of low to near-absent circulation. Consistent with this, Bayesian statistical modelling shows that the magnitude of a (sub) type's circulation in preceding seasons had only limited effect on subsequent size. Instead, the strong clustering of different countries' epidemic size within individual seasons, supported by statistical modelling, suggests that epidemic size is more strongly influenced by

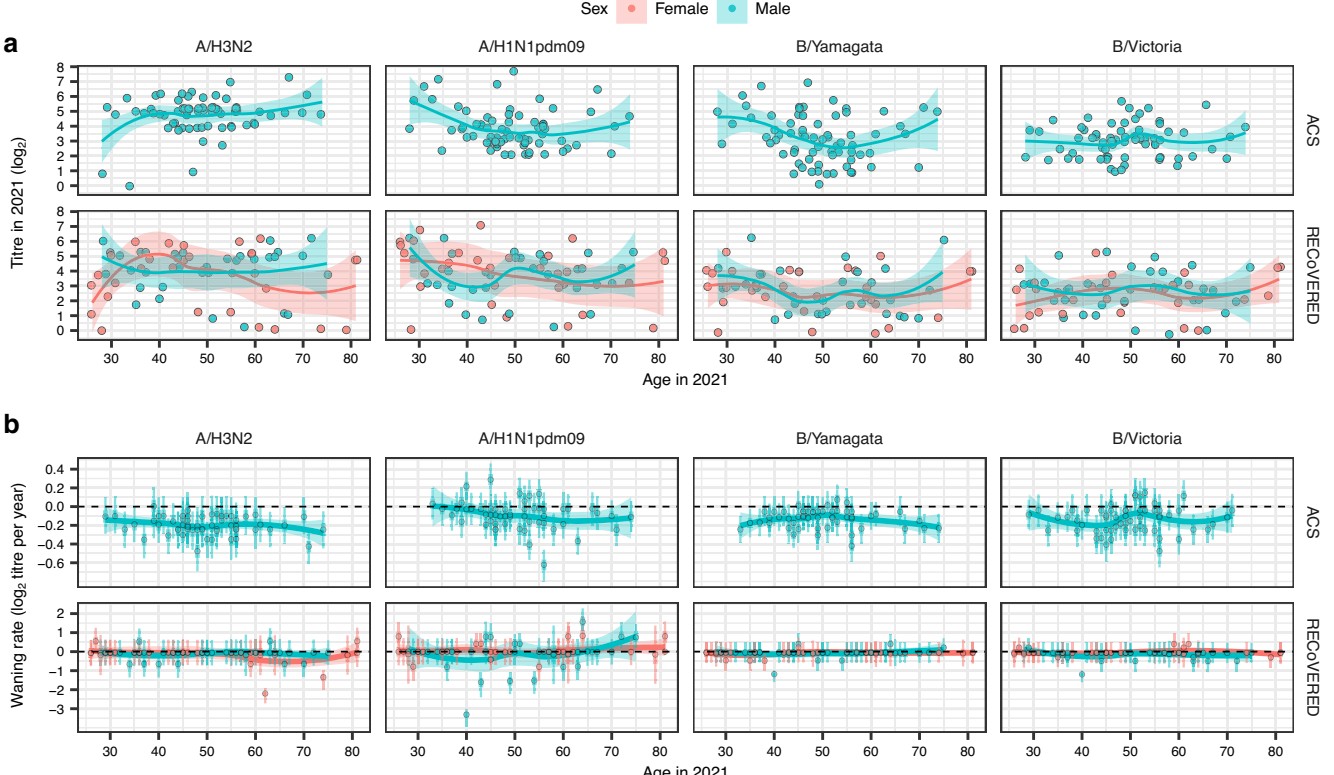

**Fig. 4 | The effects of age and sex on baseline antibody titre and waning rate. a** A cross section of antibody titres in 2021 for 70 ACS individuals and 65 RECoVERED individuals, broken down by (sub)type, age and sex. The smoothing line corresponds to a LOESS fit with span = 0.75, for each sex individually. The confidence band corresponds to a 95% confidence interval. **b** Individual-level fitted waning rates with 50% (thick lines) and 95% (narrow lines) CIs for the 2017-21 period for the individuals among the 70 ACS individuals that did not see a ≥ 4-fold increase in titre in consecutive years and the 65 RECoVERED individuals for 2020-21, broken down by (sub)type, age and sex. Points correspond to median individual-level fitted waning rates. The smoothing line corresponds to a LOESS fit with span = 0.75, for each sex individually. The confidence band corresponds to a 95% confidence interval.

season-specific effects that are unrelated to the absence or presence of circulation in the prior season(s). Similarly, the severity of influenza seasons appears to be largely independent of the magnitude of influenza virus circulation in the preceding seasons.

Given our finding that prior incidence only had a subdominant effect on subsequent epidemic sizes historically, the lack of exceptionally large or severe post-pandemic influenza seasons is not unexpected. Our results suggest that the relationship between accumulation of susceptibility and epidemic size is decidedly more complex: even if there were substantial accumulation of susceptibility during the COVID-19 pandemic, it is more likely that epidemic size and severity would be dominated by season-to-season variation due to the combined effect of other known or unknown factors unrelated to the absence or presence of substantial circulation in preceding years. Nevertheless, it is possible that periods of absent circulation much longer than seen during the COVID-19 pandemic will have substantial effects on epidemic dynamics. While the precise mechanisms are unknown and likely multifactorial, the prolonged absence of influenza virus circulation may have contributed to historical observations of severe outbreaks in remote areas, and it is possible that such instances may yield insights into the effects of longer lulls[29,30].

The precise determinants of these season effects are likely manifold, combining factors such as the flux of viral seeding, heterosubtypic competition, vaccine effectiveness, antigenic novelty, immune imprinting, and possibly other unknown factors[31-39]. While our analysis showed that an effect of prior incidence on epidemic size exists, it also shows that it is only one of likely many determinants. Crucially, our lack of understanding of many of these determinants currently limits our capacity to generate meaningful multi-year

forecasts of epidemic sizes. Predictive models that incorporate the uncertainty arising from the unpredictability of season effects will necessarily yield outputs that have wide confidence intervals, limiting their utility for public health purposes. Simultaneously, not incorporating this uncertainty will likely result in substantial prediction error. It is likely that a better understanding of the different immunological, evolutionary, ecological and epidemiological factors that determine epidemic size, beyond waning immunity, is required to perform accurate and precise multi-year prediction of epidemic size.

We also showed that HI-measured immune protection against recent seasonal influenza viruses remained largely unchanged in adults during the COVID-19 pandemic. Our analysis suggests that waning of antibody titres against seasonal influenza viruses occurs at timescales substantially longer than the lull in seasonal influenza virus circulation during the COVID-19 pandemic[10]. We showed that waning rates following periods of absent circulation were largely in agreement with waning rates previously reported for adults during regular periods of influenza virus circulation[40]. This lower waning rate is also more consistent with individual-level estimates of the duration of protection against infection by circulating strains[41], and is lower than was assumed in models used to project post-COVID-19 pandemic epidemic sizes[12,13,15]. In modelling studies, one of the reasons for the assumption of a high rate of waning is to yield annually recurring epidemics in SIR-type models[15]. In reality, epidemics recur annually with a much lower waning rate. While much is unknown about the mechanisms behind the annual recurrence of influenza epidemics, network effects might form a plausible explanation for the annual recurrence of influenza epidemics despite a limited rate of waning of protection. However, such dynamics cannot be incorporated in SIR-type compartmental

models. Hence, the inherent shortcomings of oft-used epidemiological models in modelling a relatively complex pathogen lead to the use of unrealistic parameter values, potentially leading to unreliable inferences. The complexity of influenza epidemiology and its contravention of SIR-predicted dynamics is also reflected in the empirical observation that novel antigenic variants do not necessarily lead to larger epidemic sizes[31].

Our study has some limitations. First, we used influenza-like illness data from the WHO FluID database and virologically confirmed data from the WHO FluNet database in our epidemiological analyses. Bias might affect both data sources. In particular, the FluID ILI data is not influenza-specific and might be affected by year-to-year variation in sentinel sites, and FluNet data might be biased due to the presence of convenience samples and overrepresentation of outpatient surveillance. However, the observed consistency in the estimated (sub)type-specific epidemic sizes across the 20 countries included in the analysis for any given season suggests that these data sources broadly capture influenza epidemiological dynamics. Therefore, our results are unlikely to be substantially affected by potential year-on-year differences in reporting behaviour or unrepresentative sampling. Second, the applicability of our analysis to the post-COVID-19 pandemic-like situation is predicated on the absence of heterosubtypic immunity. Importantly, heterosubtypic protection has previously been estimated to be exceedingly short-lived, with duration on the order of a single day[41]. Third, we could only perform our severity analysis based on Europe-wide excess mortality data. Nonetheless, the clustering of epidemic sizes across different European countries within a particular season as observed in our epidemiological analysis suggests that, for any given season, Europe-wide severity data is likely representative at the individual country level.

Fourth, the serum samples were collected in two independent cohorts, with substantial diversity in age and sex, accounting for the elderly but excluding children. Due to the complex effects of immunosenescence, the elderly potentially exhibit differing antibody dynamics. Whilst studies have shown that vaccine-mediated protection wanes modestly quicker in those over 65 years of age[42,43], there is little evidence to support the notion that serum antibodies wane significantly faster for this age group. Our results showed similar antibody baseline titres and waning rates for adults below and above 65 years of age, suggesting that serum antibodies in both subgroups wane at similar rates.

Due to the lack of children in our serological analysis, the extent to which their waning rates changed during the COVID-19 pandemic remains uncertain. Immune dynamics in children are known to differ from those in adults[41], with potentially higher waning rates. This could lead to increased susceptibility to infection, and the duration of protection against infection is known to be shorter in children than in adults[41,44]. Furthermore, the accrual of additional birth cohorts during prolonged periods of absence of influenza virus circulation might affect epidemic dynamics. However, the same dynamics of waning in children and population turnover also occurred in pre-pandemic (sub)type lulls. Although we could not perform the serological analysis for children, the epidemiological data we use to estimate the effects of (sub)type lulls on subsequent epidemic size does incorporate dynamics of waning in children and population turnover. For this reason, the absence of child sera is unlikely to bias our conclusions.

Finally, although participants in the RECoVERED cohort were confirmed to be unvaccinated during the study period, vaccination status for the ACS cohort was not known. However, in the Netherlands individuals <60 years of age are only eligible for influenza vaccination if they have underlying health conditions, and assuming population-wide rates of influenza vaccine uptake in the Netherlands, only 3% of the ACS individuals, who importantly were all HIV-seronegative, would be expected to be vaccinated. Combined with the similarity in antibody baseline titres and waning rates when comparing adults below

and above 60 years of age, it is unlikely the lack of vaccination status for the ACS cohort biases our conclusions.

Leveraging multiple sources of data, our results show that predicted relationships between prior epidemic magnitude, the degree of accumulation of susceptibility, and the resultant epidemic size do not accurately reflect the complexity of influenza epidemiology. Likely, our lack of understanding of many aspects of influenza epidemiology fundamentally hampers our ability to generate meaningful forecasts of epidemic size and severity.

## Methods

### Epidemic composition data

We downloaded records of virological surveillance data from the WHO FluNet[20] database for all countries in the temperate Northern and Southern Hemisphere from 2002 until 2020, or a shorter period for a limited subset of countries. We limited the dataset to countries in temperate zones because the discrete seasonal nature of influenza epidemics in temperate zones facilitates the computation of lull durations and epidemic sizes. For each country, we retained the longest sequence of consecutive seasons in which at least 20 specimens were influenza-positive. In each season, defined as the period from the period from week 40 until week 20 for the Northern Hemisphere and the entire year for the Southern Hemisphere, we computed the proportion of all positive tests that was attributable to each of A/H3N2, pandemic A/H1N1pdm09 (from the 2009 pandemic onwards), and influenza B viruses. We did not break down influenza B viruses by lineage because in many countries influenza B viruses were not further characterised.

In many seasons, only a proportion of all influenza A virus positive tests were subtyped; in those cases, we approximated the total proportion of each subtype by assuming that the subtype of the non-subtyped influenza A virus specimens were distributed according to the relative proportions of subtyped influenza A viruses. Additionally, we required that in each country-season at least 20 positive specimens were characterised (though most counts were much higher). This resulted in a dataset of 718 season-country records over a period of 18 seasons, for 47 countries. We assigned a binary variable to each (sub)type in each season for substantial circulation; we defined substantial circulation as a (sub)type accounting for at least 20% of all detections in a country in a season. Hence, in principle, all three (sub)types considered can simultaneously substantially circulate in a single season. To avoid including effects of the COVID-19 pandemic on influenza dynamics, we truncated the 2019/2020 season at the 15th of February 2020, and to avoid including the effect of the 2009 A/H1N1pdm09 pandemic, we truncated the 2008/2009 influenza season at the 1st of April 2009.

### Epidemic size data

To estimate epidemic sizes for each (sub)type, we extracted weekly records of influenza-like illness from the WHO FluID[21] database. We limited this dataset to countries for which influenza-like illness (ILI) records were available for all seasons from 2010/2011 until 2019/2020, and for which virological surveillance data was available, as described above. This period was chosen because the availability of ILI data was insufficient for the years preceding 2010. We required ILI curves to follow the expected shape of an influenza epidemic curve, i.e. peaking in winter and only sporadic isolation outside this period, and without periods of missing data. To facilitate the estimation of season effects, we only considered countries located in the Northern hemisphere. This yielded a set of 20 countries, each with 10 seasons worth of ILI data, located in Europe and the Middle East.

To approximate the relative epidemic size of each subtype in each country in each season, we first multiplied total ILI incidence in that country's season by the proportion of all detections in that country in that season attributable to that (sub)type in the virological surveillance data, yielding a measure of (sub)type-specific ILI. We then computed the relative size of each (sub)type's epidemic in each country by season

by computing the proportion of all ILI in that country in the total study period, i.e. from the 2010/2011 until 2019/2020 seasons, that was attributable to that (sub)type and season, and multiplying this number by the total number of seasons. Hence, if in a particular season in a particular country a (sub)type's epidemic had a relative size of 0.8, its size corresponded to 80% of the mean influenza epidemic size (irrespective of (sub)type) in that country in the ten-year period. In this way, this metric accounts for differences between seasons with regard to epidemic size, as opposed to only composition. We accounted for the COVID-19 pandemic as described above.

## Statistical modelling

We used Bayesian hierarchical linear regression to estimate the effects of lull periods on epidemic size. As predictors, we separately used each of (1) seasons since previous substantial circulation, (2) previous season relative size, or (3) previous two seasons' relative size sum. The first model has relative size as outcome and seasons since substantial circulation as calculated using the virological surveillance data as predictor $(n = 188, 198, 191$ for A/H3N2, A/H1N1pdm09 and B):

$$y_i \sim Normal(\alpha + \beta x_i, \sigma_y) \tag{1}$$

where $y_i$ is an epidemic's relative size for a certain (sub)type in country-season pair $i$, $\alpha$ is the model intercept for that (sub)type, $\beta$ is the coefficient for number of seasons since substantial circulation of the (sub)type, and $\sigma_y$ is the error standard deviation. $x_i$ represents the number of seasons since substantial circulation of the (sub)type in country-season pair $i$, such that $\alpha$ represents the predicted size if the previous substantial circulation was in the previous season. We put weakly informative priors on the main effect, the intercept and the standard deviation.

$$\beta \sim Normal(0,1) \tag{2}$$

$$\alpha \sim Normal(0.5,1) \tag{3}$$

$$\sigma_y \sim Half - Normal(0,1) \tag{4}$$

We also ran the same models with season effects, where we furnished each season with its own intercept:

$$y_i \sim Normal(\alpha_{s[i]} + \beta x_i, \sigma_y) \tag{5}$$

Here, $\alpha_{s[i]}$ is the season effect for that (sub)type corresponding to that season, for a country-season pair $i$. Season effects are shared between countries in a single season. We assumed that the season effects $\alpha_s$, constrained to positive values, are distributed according to a common mean $\mu_\alpha$ and common standard deviation $\sigma_\alpha$, and we put weakly informative priors on the mean season effect and its standard deviation:

$$\alpha_s \sim Normal(\mu_\alpha, \sigma_\alpha) \tag{6}$$

$$\mu_\alpha \sim Normal(0.5,1) \tag{7}$$

$$\sigma_\alpha \sim Half - Normal(0,1) \tag{8}$$

In addition to the model described above, we ran models with relative size as outcome and relative size in the previous season as the predictor $(n = 180$ for each (sub)type) or the sum of the two previous seasons' sizes $(n = 160$ for each (sub)type). We used the same model specification and priors as for the size-seasons since substantial circulation model, but we replaced the predictor with the relative size in

the previous season, or with the sum of relative size in the two previous seasons. All the above models were run both with and without season effects, i.e. with either a single value for the intercept, or with a separate intercept for each season. In the models with season effects, the season effects correspond to the predicted 'base size' of a (sub)type's epidemic in a particular season, given that either the previous substantial circulation was in the previous season, there was no circulation in the previous season, or there was no circulation in the previous two seasons, respectively, for the models with seasons since previous substantial circulation, previous season size, and previous two seasons' size sum as predictors. We ran the models for each (sub)type individually, and for each predictor individually. The models were fit using MCMC in Stan v2.21.0. The models were each run for 3000 iterations, discarding the first 1000 as burn-in, with four independent chains. Convergence was assessed by inspection of Rhat (<1.05), effective sample size (> 200) and the trace plots. We compared models with and without season effects using leave-one-out cross-validation[45]. Analyses were performed using R v4.0.3.

## Severity data

We used excess deaths as a proxy for epidemic severity. We extracted pooled Europe-wide winter-period excess mortality for the 2010/2011 and 2011/2012 seasons, and pooled-Europe wide flu-specific winter-period excess mortality for the 2012/2013 to 2017/2018 seasons from the EuroMOMO network, which combines data regarding excess deaths in European countries to estimate Europe-wide excess mortality[22,23]. The number of countries included in the calculation of excess deaths ranged from 7 in 2010/2011 to 24 in 2017/2018. For seasons that were (1) dominated by a single (sub)type in most countries; and (2) were uniform across most countries with regard to the number of seasons since previous substantial circulation of the dominant (sub)type, we compared the Europe-wide excess mortality to the value for seasons since substantial circulation. Because these criteria were only met for a small number of seasons, the comparison was performed mostly qualitatively.

## Viruses

To select the four representative strains used in this study (A/Netherlands/04189/2017 (A/H3N2), A/Netherlands/10218/2018 (A/H1N1pdm 09), B/Netherlands/04136/2017 (B/Yamagata), and B/Netherlands/00302/2018 (B/Victoria)), we downloaded high-quality (<5% ambiguous nucleotides, >95% full length) seasonal influenza virus haemagglutinin sequences (A/H3N2, $n = 1396$; A/H1N1pdm09, $n = 1283$; B/Yamagata, $n = 1129$; and B/Victoria, $n = 1408$) collected between 2016 and October 2021 from GISAID (www.gisaid.org) and reconstructed maximum-likelihood phylogenetic trees for each influenza virus subtype using the general time reversible substitution model with IQ-TREE[46]. These trees were used to assess the representativeness of viruses from the Netherlands in the early portion of the study period and the selected viruses were all representative of viruses that caused epidemics in the Netherlands during the 2017/2018 winter.

All four viruses were propagated in Madin-Darby Canine Kidney (MDCK) cells in infection medium which consisted of MEM-Eagle Medium /EBSS (Lonza, Geleen, The Netherlands) supplemented with MEM Non-Essential Amino Acids (Gibco, ThermoFischer Scientific, Amsterdam, The Netherlands), penicillin (100 IU/mL), streptomycin (100 mg/mL), L-Glutamine (Lonza), HEPES (Lonza), and TPCKtrypsin (Sigma-Aldrich/Merck, Darmstadt, Germany). They were harvested after 72 h of incubation at either 37 °C (H3N2 and H1N1) or 33 °C (Yamagata and Victoria) and checked by Sanger sequencing.

## Longitudinal serum samples

A total of 630 serum samples from 165 healthy male and female adults, including people >70 years of age (elderly), were collected in the Netherlands, longitudinally, before and during the COVID-19

pandemic in two separate cohorts: 1. the Viro-immunological, clinical and psychosocial correlates of disease severity and long-term outcomes of infection in SARS-CoV-2 – a prospective cohort study[27] (RECoVERED) and 2. Amsterdam Cohort Studies on HIV infection and AIDS[28] (ACS).

The aim of the RECoVERED cohort study is to describe the immunological, clinical and psychosocial sequelae of a SARS-CoV-2 infection. Individuals aged 16 to 85 years with laboratory-confirmed SARS-CoV-2 infection were enrolled from May 2020 until the end of June 2021 in the municipal region of Amsterdam, the Netherlands. All participants provided written informed consent. The RECoVERED study was approved by the medical ethical review board of the Amsterdam University Medical Centre (NL73759.018.20). From the RECoVERED study, we selected a total of 34 male and 31 female adults ranging from 20 to 77 years old at the time of sample collection in mid-2020, all of which had a confirmed SARS-CoV-2 infection but were otherwise healthy and unvaccinated for influenza in 2020. For these 65 individuals, samples were collected in the summer period of 2020 and 2021 only (two total for each participant).

The initial aim of the Amsterdam Cohort Studies was to investigate the prevalence, incidence, and risk factors of HIV-1 infection. The study population consists of men who have sex with men and live mainly around the city of Amsterdam, the Netherlands. Participation in ACS is voluntary and without incentive. Written informed consent of each participant was obtained at enrolment. The Amsterdam Cohort Studies on HIV infection and AIDS was approved by the Medical Ethics Committee of the Amsterdam University Medical Centre of the University of Amsterdam, the Netherlands (MEC 07/182). Participants from the ACS cohort included in our study were all HIV-1 seronegative men ranging from 22 to 70 years old at the time of sample collection in mid-2017. Briefly, five stored samples were used per participant, i.e. 1. mid-2017, 2. mid-2018, 3. mid-2019, 4. mid-2020, 5. mid-2021.

## Haemagglutination inhibition (HI) assay

All serum samples were receptor destroying enzyme (RDE)-treated, as described elsewhere[47]. Briefly, for ACS individuals 1–30, 100 µL of serum was combined with 200 µL of RDE (Denka Seiken, Tokyo, Japan); for ACS individuals 31–100, 100 µL of serum were combined with 300 µL or 200 µL of RDE (supplied by the National Institute for Public Health and the Environment). This difference in protocol was per the instructions of the providers of the respective batches of RDE. For all 65 RECoVERED subjects, the latter RDE was used, combining 100 µL of serum with combined with 200 µL of RDE. Because of this protocol difference, the results of ACS participants 1–30 and 31–100 are shown separately. Differences in dilution were accounted for in titre calculation. All samples were then incubated at 37 °C for 18–20 h. The RDE reaction was then halted by heating the treated samples at 56 °C for 30–60 min.

The haemagglutination inhibition activity of all serum samples was tested in an HI assay as described elsewhere[47] using two replicates per sample for A/H1N1pdm09, B/Yamagata, and B/Victoria viruses, and one single measurement for A/H3N2 viruses. Due to inefficient agglutination of turkey red blood cells (tRBCs) by recent A/H3N2 viruses, we used glycan remodelled tRBCs expressing appropriate receptors for recent A/H3N2 viruses for the HI assays of the A/H3N2 virus stock[48]. Briefly, the haemagglutination titre of each of the four viruses was determined by performing a two-fold serial dilution of 50 µL of each virus stock and adding 50 µL of PBS and 25 µL of 1% turkey red blood cells (tRBCs) to each well, followed by 1 h incubation at 4 °C and the reading of the haemagglutination patterns. The virus stocks were then diluted to a concentration of four haemagglutination units (HAU). The diluted viruses were then incubated with 50 µL of 2-fold serially diluted serum, in a total volume of 75 µL for 30 min at 37 °C. The initial dilution used for the serial dilution of the serum was 1:20 of the RDE-treated serum. After the incubation step, 25 µL of 1% turkey red blood cells

were added to the serum-virus mix and incubated at 4 °C for 1 h. The haemagglutination inhibition patterns were then read out and used for the calculation of antibody titres.

## Antibody waning model

For the RECoVERED cohort for the years 2020 and 2021, all participants were confirmed to have not received an influenza vaccination between the two sample collections and no natural influenza infection can be safely assumed given the near absence of influenza in the Netherlands during this period. For the ACS data, those who experienced a four or greater fold increase in titre between consecutive visits for a particular strain had their strain-data discarded in order to remove the obscuring effects of vaccination and infection.

True antibody titre $\log_2$ HI, $\widetilde{T}_i$ as opposed to that measured by HI assay, $T_i$, is a continuous variable which we assume, for every individual $i$, decays with time $t$ as

$$\widetilde{T}_i = c_i - \alpha_i t \tag{8}$$

Where $c_i$ are individual specific initial titres and $\alpha_i$ are the individual waning rates. The waning rates are assumed to be normally distributed about a population mean, $\alpha_\mu$, with standard deviation, $\alpha_\sigma$.

If serum dilutions could be performed in arbitrarily small increments, we assume the point at which haemagglutination would be observed to cease, $T_{obs}$, to be distributed normally about the true value:

$$T_{obs} \sim N(\widetilde{T}, \epsilon) \tag{9}$$

Instead, with discrete dilutions in increments of one, the probability of measuring $T \in \{0,1,2...8\}$ is the probability that $T_{obs}$ falls between $T$ and $T-1$. Thus, the measurement probability is given by:

$$P(T|\widetilde{T},\epsilon) = \begin{cases} \Phi(1,\widetilde{T},\epsilon) & T < 1 \\ \Phi(T,\widetilde{T},\epsilon) - \Phi(T-1,\widetilde{T},\epsilon) & 1 \le T < 8 \\ 1 - \Phi(8,\widetilde{T},\epsilon) & T \ge 8 \end{cases} \tag{10}$$

where $\Phi(x, \mu, \sigma)$ is the cumulative distribution function of the normal distribution.

Our data for each individual, $i$, consists of a series of titre measurements, $\mathbf{T}_{i,r} = (T_{1,i,r}, T_{2,i,r}, \ldots, T_{n,i,r})$, at corresponding timepoints $1 \ldots n$, where $r \in \{1,2\}$ indicates replicate measurements. To infer the probability of the unknown parameters $\epsilon$, $\alpha_\mu$ and $\alpha_\sigma$ given the data, it is necessary to augment the data by introducing individual intercepts. For one replicate from one individual, the likelihood of unknown parameters $\alpha_\mu$, $\alpha_\sigma$, $\epsilon$, and $c_i$ then becomes:

$$p(\alpha_\mu,\alpha_\sigma,\epsilon,c_i|\mathbf{T}_{i,r}) \propto p(\mathbf{T}_{i,r}|\alpha_\mu,\alpha_\sigma,\epsilon,c_i)\Pi(\alpha_\mu,\alpha_\sigma,\epsilon,c_i) \tag{11}$$

$$\propto p(\mathbf{T}_{i,r}|\widetilde{\mathbf{T}}_i(\alpha_\mu,\alpha_\sigma,\epsilon,c_i))\Pi(\alpha_\mu,\alpha_\sigma,\epsilon,c_i) \tag{12}$$

where $\widetilde{\mathbf{T}}_i(\alpha_\mu,\alpha_\sigma,\epsilon,c_i) = (\widetilde{T}_{1,i}, \widetilde{T}_{2,i}, \ldots, \widetilde{T}_{n,i})$ are the true values of titre, given the unknown parameters and $\Pi$ is the prior joint distribution of the parameters. The total log likelihood is thus the sum over all individuals and replicates:

$$L(\alpha_\mu,\alpha_\sigma,\epsilon,c|\mathbf{T}) \propto \sum_i \sum_r \log(p(\alpha_\mu,\alpha_\sigma,\epsilon,c_i|\mathbf{T}_{i,r})) \tag{13}$$

A Markov Chain Monte Carlo (MCMC) algorithm implemented in Stan v2.21.0 was used to explore the distribution of model parameters (waning and measurement) and augmented data (individual intercepts). This model was run on four independent chains, each

consisting of 5000 iterations discarding the first 2500 as burn in. Weakly informative priors were used and convergence was assessed by inspection of the trace plots and Rhat. Analyses were conducted using R v4.0.3, with code available in the GitHub repository.

## Reporting summary

Further information on research design is available in the Nature Portfolio Reporting Summary linked to this article.

## Data availability

Raw surveillance data is available from WHO FluNet (https://www.who.int/tools/flunet) and FluID (https://www.who.int/teams/global-influenza-programme/surveillance-and-monitoring/fluid). Accession codes for GISAID data are provided in Supplementary Data 1. Biological materials are available for study via the Amsterdam Cohort Studies on HIV infection and AIDS (ACS) and the Viro-immunological, clinical and psychosocial correlates of disease severity and long-term outcomes of infection in SARS-CoV-2 – a prospective cohort study (RECoVERED). Processed data is provided in the source data file and available at the project GitHub repository (https://github.com/AMC-LAEB/waning-immunity-to-flu). Source data are provided with this paper.

## Code availability

Custom scripts used for data analysis and modelling are available at the project GitHub repository (https://github.com/AMC-LAEB/waning-immunity-to-flu) and at https://doi.org/10.5281/zenodo.10276853.

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

## Acknowledgements

A.X.H., Z.C.F.G. and C.A.R. were supported by ERC NaviFlu (No. 818353). J.G. and C.A.R. were supported by NIH R01 (5R01AI132362-04). C.A.R. was also supported by an NWO Vici Award (09150182010027). R.p.d.V. was supported by ERC starting grant 802780 and a Beijerinck Premium of the Royal Dutch Academy of Sciences. G.J.B. was supported by the Netherlands Organization for Scientific Research (NWO TOPPUNT 718.015.003) and by an ERC advanced grant (101020769). The RECoVERED cohort is supported by NWO ZonMw (No. 10150062010002) and the Public Health Service of Amsterdam (Research & Development grant number 21–14). The Amsterdam Cohort Studies on HIV infection and AIDS, a collaboration between the Public Health Service Amsterdam, the Amsterdam UMC of the University of Amsterdam, Medical Center Jan van Goyen and the HIV Focus Center of the DC-Clinics, are part of the Netherlands HIV Monitoring Foundation and financially supported by the Center for Infectious Disease Control of the Netherlands National Institute for Public Health and the Environment. We gratefully acknowledge the authors and originating and submitting laboratories (supplementary information) for the reference sequences retrieved from GISAID's EpiFlu Database used in this study. The authors thank all ACS and RECoVERED study participants. We are also grateful to Mr. Reinier van der Palen of the department of Chemical Biology and Drug Discovery, Utrecht University for his practical assistance with turkey erythrocyte glycan remodelling.

## Author contributions

S.P.d.J., Z.C.F.G., J.C.G., A.X.H., D.E., M.D.d.J. and C.A.R. designed the research; Z.C.F.G, S.v.L., M.v.H., K.d.H. and L.E.v.G executed the experimental work; Z.C.F.G. and S.v.L. generated the antibody titre data; E.W., G.J.d.B, H.D.G.v.W., A.M., M.B., L.v.d.H., M.P., N.K. and M.D.d.J. collected the clinical samples; R.P.d.V. and G.J.B. made and provided the glycan remodelled turkey red blood cells; S.P.d.J. and J.C.G. implemented the modelling work and performed the data analysis; S.P.d.J., Z.C.F.G., J.C.G., A.X.H., K.D.H., B.E.N. and C.A.R. wrote the first draft of the paper. All authors contributed to the critical revision of the paper. Z.C.F.G. and J.C.G. contributed equally.

## Competing interests

The authors declare no competing interests.
