## [Peer Review File · Nature Communications]

Determinants of epidemic size and the impacts of lulls in seasonal influenza virus circulationReviewers' comments:

Reviewer #1 (Remarks to the Author):

Review of "Impacts of the COVID-19 pandemic on future seasonal influenza epidemics" by Zandra C. Felix Garza et al.

The authors assess the likely size of the influenza epidemics in the seasons following the big corona lock-downs. The authors present two empirical findings.

Firstly they estimate waning of immunity as measured by the decrease in anti-body titre among a group of adult males. This data is interesting because it has been collected during the corona-period where influenza has been absent such that the anti-body dynamics reflect the true waning in the absence of reinfection or boosting by exposure to infection. For all 4 lineages (H3n2, H1N1, B/Victoria and B/Yamagata) they find decay-rates corresponding the half-times of 3.5-10 years, consistent with the literature.

The second set of data concerns the season to season dynamics of the regular winter epidemics. Specifically the authors determine how the probability that a new epidemic of the same lineage will arise in the subsequent seasons depends on the size of the epidemic in the first season -- and likewise how the size of the subsequent epidemic depends on the size of the initial epidemic. The authors find that there is indeed such a connection between epidemic size and probability of outbreak and size in subsequent seasons. However by comparing the dynamics in different areas of Europe, they find that the main effect is due to heterogeneity in viral properties (the result of anti-genic drift) rather than the epidemic size it self. This is a very interesting and important analysis that sheds new light of influenza drift-evolution and the epidemiology of winter-influenza..

The authors now combine their two finding to suggest that a) the exact level of preexisting immunity in the host-population has only limited influence of on epidemic size and that b) immunity wanes so slowly that two extra season with no influenza epidemic does not affect the size of the next epidemic -- indicating that we should not expect the first influenza epidemics after the lock-downs to be exceptionally large.

Thus the authors address an important and very timely question. However I don't believe that their line of reasoning holds for two reasons.

1) immunity loss over two seasons is larger that the authors indicate: an additional 2-3 birth cohorts enter the host population and for influenza we know that these young cohorts play a substantial role in disease transmission -- high contact rates in schools play a big role for flu (though perhaps not for corona) this gives a substantial contribution oto disease transmission. While waning for adults may be as described, this is not the case for the elderly part of the population where immunity loss is much faster. So the immuno-dynamics of the population is not very well characterized by the their study.

2) The authors seem to think of the size of the seasonal epidemic is the prerequisite for the start of the next epidemic. I would suggest that the season to season dynamic should rather be interpreted as the result of influenza drift-dynamics such that when the drift dynamics are disturbed, we should no loger expect to see this pattern.

Of course RSV is a very different virus, but the unseasonal and unusually large RSV-epidemic that was observed in many European countries in the fall of 2021, also suggests recurrent epidemics of respiratory infections are substanttially affected by the lock-downs.

Reviewer #2 (Remarks to the Author):

This authors analyse longitudinal HI data from the Netherlands and population level influenza dynamics globally to assess the potential impact of suppression of influenza during COVID-19 on future influenza epidemics. This is an important question, with some useful novel data presented,

but I was not convinced that the range of results shown sufficiently supported the main conclusion currently being drawn (i.e. 'the sizes of future seasonal influenza virus epidemics will likely be similar to those observed before the pandemic'). In particular, there are multiple reasons that an interruption in influenza transmission during 2020-21 could affect subsequent dynamics, including effect on antigenic evolution and higher than usual population turnover in the inter-epidemic period, which are not discussed. In the manuscript, analysis of future epidemic size focuses almost entirely around the impact of waning influenza immunity, and the potential interaction of this process with 2020-21 suppression, so it would be helpful to make this focus clearer and have more discussion of its limitations.

Main comments:

- As an extreme example to illustrate the potential role of population turnover, suppose we performed a similar analysis on pre-vaccination measles dynamics in a population where transmission had been interrupted for two years. Serology would not suggest that antibody levels had waned (e.g. Antia et al. PLOS Biol 2018), and prior epidemic data would indicate periods of low incidence, with substantial variation in year-on-year epidemic size. However, having an additional two years of new births without any accumulated immunity within younger groups could lead to a very different dynamic once measles transmission restarted. Although influenza transmissibility is much lower, it is worth addressing why we would not expect a similar dynamic for influenza in the coming years, given work showing the role of children in driving flu epidemics (e.g. Baguelin et al. PLOS Med 2013).

- It would be worth having more discussion of the concept of set point titres when introducing waning. For example, the authors cite Fonville et al. among others to support a claim that the half life is 3.5-10 years, but Fonville et al. actually state 'the broad initial response was followed by a period of titer decay during which antibody titers stabilized to form an altered antibody landscape over the course of ~1 year' and 'there is little evidence of further decay for samples beyond approximately a year post-infection, in line with the findings by Horsfall et al'. If waning occurs predominantly within season before settling on a set point value, we'd expect transient HI decline from a peak to have limited effect on future seasons a priori?

- The paper seemed to end quite abruptly after the main results. I would have liked to see more discussion of the wider contributing factors to influenza dynamics mentioned above, as well as more discussion of the limitations of the current analysis and what future data might be required to provide further insights (e.g. are there particular 'sentinel' countries or age groups we should be looking at?) At present, I don't think there is sufficient evidence presented to support conclusions such as 'The size of future influenza epidemics is likely to fall within the size distribution of epidemics in the years before the COVID-19 pandemic.'

- The introduction states that annual influenza epidemics infect 5-15% of populations. This seems low given subsequent evidence from in-depth sampling studies? (e.g. Cohen et al, Lancet Global Health, 2021)

Reviewer #3 (Remarks to the Author):

The paper has attempted to use residual sera from an existing longitudinal study of men and notifications to the WHO flunet database to draw conclusions about the likely level of susceptibility to influenza in 2022 given the absence of disease for 2 years. The paper is a naïve analysis of a complex immunological situation, ignores key risk groups and does not adequately address the limitations of its data sources.

1. The paper states the sera were tested against representative viruses. However, it is unclear what these viruses were representative of. Are they the vaccine strains? I could not find a list of the specific strains characterised in the manuscript. Moreover, it is unclear what year these viruses are supposed to represent; 2019? Or 2022? Finally, were these cell or egg-grown antigens? The authors' conclusions about their serology might be quite different, depending on the choice of antigen.

2. It seems like only a single antigen has been assessed for each subtype or lineage. This ignores

the fact that multiple antigenically-distinct viruses are currently in circulation. So statements about population immunity are misleading.

3. That naturally-acquired antibodies do not wane beyond a certain point is already known. There are longitudinal cohorts that have shown this (e.g. DOI: 10.1093/infdis/jiaa293). But that doesn't mean that people are not susceptible to reinfection with antigenically distinct viruses. Hence, the study's serological findings are not terribly novel or meaningful without a broader representation of circulating antigens.

4. The sample used for serological analysis consists of 165 men. This sample is not representative of sex, age, race or geography.

a. Immunological differences are known to occur between men and women (Sabra Klein has written on this topic extensively). Ignoring half the population is highly remiss and not well justified.

b. Of greater concern is that this sample does not adequately represent the age distribution of the population. It ignores older adults, who are the most vulnerable to severe outcomes of influenza disease, and who may additionally frail due to extended periods in isolation as part of pandemic mitigation measures. It also, and crucially, ignores children, many millions of whom have been born during the past 2 years and are entirely susceptible. Children 2 years have high hospitalisation rates, which may be exacerbated if the pool of susceptibles is larger than usual which could place significant stress on healthcare systems. Although the cohorts available to the researchers did not permit assessment of waning in children, cross-sectional cohorts using residual sera could have provided some information for this age group. Perhaps the best age group to study to make inferences about potential population susceptibility would really be school-aged children, for whom social mixing patterns play a key role in transmission.

c. The serological sample is not representative globally. Influenza has continued to circulate in some regions, notable equatorial regions. These serological observations are unlikely to be applicable to people living in those regions.

5. The analysis of flunet data has several limitations.

a. There doesn't seem to have been any sampling from tropical regions, which have demonstrated their importance for influenza circulation during the COVID-19 pandemic. Moreover, circulation is not necessarily seasonal in those regions.

b. Flunet is not representative of what happens in a country. Large countries with a single NIC may not be nationally representative. Also, reporting is sometime interrupted, which could falsely indicate a lull in activity.

c. Grouping influenza B lineages ignores that these lineages are antigenically distinct.

6. The conclusions ignore evidence from other viruses, such as RSV, that disruptions to usual seasonality has led to "rebound" epidemics of greater intensity and hospitalisation burden than usual; and that epidemics of influenza are unlikely to occur during their "usual" seasonal period.

Disruptions to usual seasonality could continue for several years (e.g. DOI: 10.1073/pnas.2013182117). That alone presents a problem for the healthcare system which times vaccination programmes and surge capacity for winter epidemics.

Reviewer #4 (Remarks to the Author):

Review of Garza et al, "Impacts of the COVID-19 pandemic on future seasonal influenza epidemics" for Nature Communications.

1. What are the noteworthy results?

The authors set out to answer a most pressing question: will seasonal influenza return more deadly than before, due to its absence during the COVID-19 pandemic period. Based on serology data from healthy adults (a population from an Amsterdam HIV study), the authors find convincingly that the Ab levels in this population did not reduce significantly while influenza was gone, in the 2-year pandemic period. They also review past examples of lulls in influenza virus circulation.

They then conclude, based on these findings, that influenza will be no more severe than previously, nothing to worry about.

But while the authors succeed in demonstrating that Ab levels are not significantly lower among healthy adults, I do not agree with the conclusion. This is because the study does not consider

children, the driver of influenza epidemics. Nor does it consider Ab waning among elderly, those who are most likely to suffer severe influenza or even die.

Consider the amount of serious illness/deaths is given by this simple relationship: number of infected x probability of serious illness given infection. This has not been given consideration in this paper. Here is why:

Children could well have Ab waning and thus higher attack rates: It is actually likely that children – who are not done building their robust immunity against influenza – will have lower Ab levels/waning, and thus the epidemic size will be far greater (driven by that age groups, school age children).

Elderly could well have Ab waning: Meanwhile, elderly, the age group that fares worst with influenza, may well have Ab waning. This is what is typically seen for vaccinated elderly, Ab wanes over a 6 month period. They are not studied in a group of healthy (younger) adults as in this study.

Taken together, higher attack rates in children spills over in other age groups. Thus, elderly are at a higher risk for getting infected. And when they do, likely have a higher risk of dying due to Ab waning. This means, one could make a good case that a near future influenza epidemic (especially of H3N2 in which the elderly typically far worst of all subtypes) will cause more infections and more serious illness.

The way to address this would be to consider the extreme age groups. How does immunity build up in children – and at what rates does it typically wane? Perhaps serology studies in children. Next, Ab waning among elderly is a critical aspect of making this argument; set an Ab study in a population over 65, or better, over 75 years of age.

2. Will the work be of significance to the field and related fields? How does it compare to the established literature? If the work is not original, please provide relevant references.

The question is of course of relevance to us all – so also to people in their field and related fields, like epidemiology, economics and policy. However, the conclusion is not relevant, unless the concerns above are addressed: will there be higher attack rates among children due to Ab waning? Will elderly have a higher risk of severe influenza due to Ab waning?

3. Does the work support the conclusions and claims, or is additional evidence needed?

No. The study does not support not the claim that seasonal influenza after the pandemic period will not be severe. See my rationale above.

4. Are there any flaws in the data analysis, interpretation and conclusions? - Do these prohibit publication or require revision?

Yes, it is a problem that children and elderly are not specifically studied.

It is also not clear whether a HIV diagnosis in the study population is interfering with the Ab level findings? Would be preferable that the study population is the control population in the Amsterdam HIV study (if there is one).

Finally, the study does not discuss particularly remarkable events of "lulls", in which seasonal influenza in fact was far more severe after a long absence. A good example is the deadly influenza H3N2 epidemic in Madagascar that occurred in 2002, after a long absence of this subtype from the island, see here: (<https://pubmed.ncbi.nlm.nih.gov/12458917/>).

5. Is the methodology sound? Does the work meet the expected standards in your field?

Yes the study is well done for the healthy adult study population, and review of past epidemic lulls. But serology data for extreme age groups (children, elderly) are sorely missing.

6. Is there enough detail provided in the methods for the work to be reproduced?

Yes

Our manuscript was originally transferred to Nature Communications in early 2022 while still formatted for Nature. It was reviewed by 4 reviewers (see below). There have been two major changes since that submission:

1. The primary premise of the manuscript is different. The original submission sought to predict the impact of the COVID-19 pandemic on future influenza virus circulation, concluding that post-pandemic epidemics were unlikely to be unusually large or severe compared to pre-pandemic epidemics. We are resubmitting our manuscript in the post-pandemic era where seasonal influenza virus epidemics have generally not been unusually large or severe. Our manuscript recasts our original evidence as a partial explanation of the observed reality.
2. We have substantially refocused, revised, and reformatted our manuscript to account for the fundamentally changed premise of the paper, address the reviewers' comments, and to take advantage of the space afforded to submissions at Nature Communications. Our manuscript now includes substantially more context and discussion, which also makes it a broader discussion of the complexities of influenza virus epidemiology.

Reviewers' comments: Author Responses

Reviewer #1 (Remarks to the Author):

Review of "Impacts of the COVID-19 pandemic on future seasonal influenza epidemics" by Zandra C. Felix Garza et al.

The authors assess the likely size of the influenza epidemics in the seasons following the big corona lock-downs. The authors present two empirical findings.

Firstly they estimate waning of immunity as measured by the decrease in anti-body titre among a group of adult males. This data is interesting because it has been collected during the corona-period where influenza has been absent such that the anti-body dynamics reflect the true waning in the absence of reinfection or boosting by exposure to infection. For all 4 lineages (H3n2, H1N1, B/Victoria and B/Yamagata) they find decay-rates corresponding the half-times of 3.5-10 years, consistent with the literature.

The second set of data concerns the season to season dynamics of the regular winter epidemics. Specifically the authors determine how the probability that a new epidemic of the same lineage will arise in the subsequent seasons depends on the size of the epidemic in the first season -- and likewise how the size of the subsequent epidemic depends on the size of the initial epidemic. The authors find that there is indeed such a connection between epidemic size and probability of outbreak and size in subsequent seasons. However by comparing the dynamics in different areas of Europe, they find that the main effect is due to heterogeneity in viral properties (the result of anti-genic drift) rather than the epidemic size it self. This is a very interesting and important analysis that sheds new light of influenza drift-evolution and the epidemiology of winter-influenza..

The authors now combine their two findings to suggest that a) the exact level of preexisting immunity in the host-population has only limited influence on epidemic size and that b) immunity wanes so slowly that two extra seasons with no influenza epidemic does not affect the size of the next epidemic -- indicating that we should not expect the first influenza epidemics after the lock-downs to be exceptionally large.

Thus the authors address an important and very timely question. However I don't believe that their line of reasoning holds for two reasons.

1) immunity loss over two seasons is larger than the authors indicate: an additional 2-3 birth cohorts enter the host population and for influenza we know that these young cohorts play a substantial role in disease transmission -- high contact rates in schools play a big role for flu (though perhaps not for corona) this gives a substantial contribution to disease transmission. While waning for adults may be as described, this is not the case for the elderly part of the population where immunity loss is much faster. So the immuno-dynamics of the population is not very well characterized by their study.

We thank the reviewer for their careful consideration of our manuscript. Our study included elderly individuals and the waning rates in the elderly were highly similar to those observed in other adults. This is shown in Fig 4.

While we appreciate the concern about the lack of children in the study, immune dynamics in children are included in our epidemiological analysis: in the period prior to the COVID-19 pandemic, lulls in circulation of certain influenza subtypes would have led to the accumulation of similar birth cohorts, but our analysis shows that in this period there was only a weak effect of lull duration on epidemic size, and that epidemic size was more strongly determined by other factors. Given that our manuscript is no longer trying to predict what influenza epidemics might look like after the COVID pandemic, and is instead trying to explain why they were not especially large during the first influenza seasons following the pandemic, the evidence from previous influenza lulls is perhaps more compelling.

At the same time, our serological analysis is valuable because it shows that antibody titres waned at rates similar to those observed in the pre-pandemic period, and waned at rates lower than assumed in modelling studies that predicted substantially increased epidemic size. These results hold regardless of the absence of sera from children.

2) The authors seem to think of the size of the seasonal epidemic as the prerequisite for the start of the next epidemic. I would suggest that the season to season dynamic should rather be interpreted as the result of influenza drift-dynamics such that when the drift dynamics are disturbed, we should no longer expect to see this pattern.

We are not entirely sure what the reviewer means here. Seasonal influenza A/H3N2 and A/H1N1 viruses routinely cause epidemics over multiple years in the same locations even in the absence of antigenic change (or, in the language here, absence of drift dynamics). The rationale for why epidemics of seasonal influenza should have been bigger following the COVID-19 pandemic was predicated on the widely held assumption that small or no

influenza epidemics would lead to a loss of immunity and thus bigger epidemics following the pandemic.

Of course RSV is a very different virus, but the unseasonal and unusually large RSV-epidemic that was observed in many European countries in the fall of 2021, also suggests recurrent epidemics of respiratory infections are substantially affected by the lock-downs.

These patterns were highly heterogeneous and happened during a time of substantially enhanced surveillance infrastructure compared to the pre-pandemic period. Importantly, the influenza seasons following the pandemic have now occurred and the epidemics were generally not larger or more severe. This pattern warrants explanation, particularly given all the forecasts to the contrary during the pandemic.

Reviewer #2 (Remarks to the Author):

This authors analyse longitudinal HI data from the Netherlands and population level influenza dynamics globally to assess the potential impact of suppression of influenza during COVID-19 on future influenza epidemics. This is an important question, with some useful novel data presented, but I was not convinced that the range of results shown sufficiently supported the main conclusion currently being drawn (i.e. 'the sizes of future seasonal influenza virus epidemics will likely be similar to those observed before the pandemic'). In particular, there are multiple reasons that an interruption in influenza transmission during 2020-21 could affect subsequent dynamics, including effect on antigenic evolution and higher than usual population turnover in the inter-epidemic period, which are not discussed. In the manuscript, analysis of future epidemic size focuses almost entirely around the impact of waning influenza immunity, and the potential interaction of this process with 2020-21 suppression, so it would be helpful to make this focus clearer and have more discussion of its limitations.

We thank the reviewer for their comments. We have substantially expanded the discussion to cover these topics and have refocused our manuscript in light of the fact that the first post-pandemic influenza seasons have now occurred and were typically neither larger nor more severe than their pre-pandemic counterparts.

Main comments:

- As an extreme example to illustrate the potential role of population turnover, suppose we performed a similar analysis on pre-vaccination measles dynamics in a population where transmission had been interrupted for two years. Serology would not suggest that antibody levels had waned (e.g. Antia et al. PLOS Biol 2018), and prior epidemic data would indicate periods of low incidence, with substantial variation in year-on-year epidemic size. However, having an additional two years of new births without any accumulated immunity within younger groups could lead to a very different dynamic once measles transmission restarted. Although influenza transmissibility is much lower, it is worth addressing why we would not expect a similar dynamic for influenza in the coming years, given work showing the role of children in driving flu epidemics (e.g. Baguelin et al. PLOS Med 2013).

As with the similar comment from reviewer 1, the lack of accumulation in immunity in younger groups would have similarly applied to pre-pandemic lulls and the impact seems to be substantially outweighed by other factors. The fact that influenza virus epidemiology behaves in a different way than one would expect (and the implications of this fact for e.g. forecasting) is a key message of the new paper.

- It would be worth having more discussion of the concept of set point titres when introducing waning. For example, the authors cite Fonville et al. among others to support a claim that the half life is 3.5-10 years, but Fonville et al. actually state 'the broad initial response was followed by a period of titer decay during which antibody titers stabilized to form an altered antibody landscape over the course of ~1 year' and 'there is little evidence of further decay for samples beyond approximately a year post-infection, in line with the findings by Horsfall et al'. If waning occurs predominantly within season before settling on a set point value, we'd expect transient HI decline from a peak to have limited effect on future seasons a priori?

The reviewer hits on an important topic and one that is still subject to much research. It is the case that antibody titres do wane substantially during the first 6 months post infection. However, we find infections that did occur in our cohort in the pre-pandemic period were always in individuals with lower-than-average antibody titres that were close to the broadly used protective threshold of HI 40, suggesting that whatever waning has occurred from the previous season results in titres that are correlated with protection from infection.

- The paper seemed to end quite abruptly after the main results. I would have liked to see more discussion of the wider contributing factors to influenza dynamics mentioned above, as well as more discussion of the limitations of the current analysis and what future data might be required to provide further insights (e.g. are there particular 'sentinel' countries or age groups we should be looking at?) At present, I don't think there is sufficient evidence presented to support conclusions such as 'The size of future influenza epidemics is likely to fall within the size distribution of epidemics in the years before the COVID-19 pandemic.'

First, our manuscript is now presented as post-hoc exploration of the observation that post-pandemic influenza seasons have generally not been larger than their pre-pandemic counterparts. Second, we have substantially expanded the discussion to more thoroughly present our results and their interpretation.

- The introduction states that annual influenza epidemics infect 5-15% of populations. This seems low given subsequent evidence from in-depth sampling studies? (e.g. Cohen et al, Lancet Global Health, 2021)

We have modified the text to reflect this comment and include this reference.

Reviewer #3 (Remarks to the Author):

The paper has attempted to use residual sera from an existing longitudinal study of men and

notifications to the WHO flunet database to draw conclusions about the likely level of susceptibility to influenza in 2022 given the absence of disease for 2 years. The paper is a naïve analysis of a complex immunological situation, ignores key risk groups and does not adequately address the limitations of its data sources.

1. The paper states the sera were tested against representative viruses. However, it is unclear what these viruses were representative of. Are they the vaccine strains? I could not find a list of the specific strains characterised in the manuscript. Moreover, it is unclear what year these viruses are supposed to represent; 2019? Or 2022? Finally, were these cell or egg-grown antigens? The authors' conclusions about their serology might be quite different, depending on the choice of antigen.

The viruses were representative of the viruses that circulated globally immediately prior to the pandemic. A specific list of strains is now included in the main text. All of the viruses were cell-grown, exactly as they should be. Given that our viruses were representative of the viruses that circulated immediately before the pandemic, they should provide a reasonable picture of the immune landscape to those viruses and viruses that are up to one antigenic cluster away. To this end, there is no reason to believe that our results are biased by our choice of antigens.

2. It seems like only a single antigen has been assessed for each subtype or lineage. This ignores the fact that multiple antigenically-distinct viruses are currently in circulation. So statements about population immunity are misleading.

All of the antigenic variants that were circulating in the years immediately prior to the pandemic were antigenically cross-reactive with one another. While it would have been more extensive to include the other circulating variants of H3, given the absence of waning during the pandemic and the cross-reactivity among these variants, there is no reason to believe that the inclusion of other variants would have revealed patterns of waning not identified in this study.

3. That naturally-acquired antibodies do not wane beyond a certain point is already known. There are longitudinal cohorts that have shown this (e.g. DOI: 10.1093/infdis/jiaa293). But that doesn't mean that people are not susceptible to reinfection with antigenically distinct viruses. Hence, the study's serological findings are not terribly novel or meaningful without a broader representation of circulating antigens.

We disagree. There have been no other published longitudinal cohort studies of immunity to influenza that covered the pre-pandemic and pandemic periods. Other studies have shown this in the pre-pandemic period but numerous studies assumed that these results did not apply during the pandemic period. To this end, our study provides valuable evidence.

4. The sample used for serological analysis consists of 165 men. This sample is not representative of sex, age, race or geography.

It is correct that this was one of the cohorts included in our study. We also included a cohort that included women and the elderly – the results in this group were highly similar to the group of 165 men. We could not include children for the reasons explained in the

manuscript. We also could not include a broader geographic range, but it is not clear how this would meaningfully change the results.

a. Immunological differences are known to occur between men and women (Sabra Klein has written on this topic extensively). Ignoring half the population is highly remiss and not well justified.

There was no evidence for this in antibody titres from women in our cohort. Explicit comparisons between men and women are now shown in Fig 4.

b. Of greater concern is that this sample does not adequately represent the age distribution of the population. It ignores older adults, who are the most vulnerable to severe outcomes of influenza disease, and who may additionally frail due to extended periods in isolation as part of pandemic mitigation measures. It also, and crucially, ignores children, many millions of whom have been born during the past 2 years and are entirely susceptible. Children 2 years have high hospitalisation rates, which may be exacerbated if the pool of susceptibles is larger than usual which could place significant stress on healthcare systems. Although the cohorts available to the researchers did not permit assessment of waning in children, cross-sectional cohorts using residual sera could have provided some information for this age group. Perhaps the best age group to study to make inferences about potential population susceptibility would really be school-aged children, for whom social mixing patterns play a key role in transmission.

As mentioned above, our cohort did include elderly individuals and their patterns were highly similar to those of other adults. We agree that including children would have been valuable and interesting but there are exceedingly few longitudinal cohorts that include children and none that we have access to. Cross-sectional cohorts would be fine but would be wildly different in quality compared to our longitudinal cohorts and would be subject to different criticisms. The accumulation of birth cohorts also occurred in pre-pandemic lulls, and our analysis shows that in the pre-pandemic period the effect of lulls in circulation on epidemic size was limited, and hence our study does account for possible effects of population turnover.

c. The serological sample is not representative globally. Influenza has continued to circulate in some regions, notable equatorial regions. These serological observations are unlikely to be applicable to people living in those regions.

With a small number of possible exceptions, influenza virus circulation was at a historic low during the pandemic. Our results should be applicable to the majority of the world's countries. For countries where influenza viruses continued to circulate during the pandemic, it is not clear that our study should be relevant since it focuses on the impact of lulls in circulation. Also, epidemics in those countries would not have experienced the lull and thus the post-pandemic patterns in those countries should not be effected by the phenomena we sought to study.

5. The analysis of flunet data has several limitations.

a. There doesn't seem to have been any sampling from tropical regions, which have

demonstrated their importance for influenza circulation during the COVID-19 pandemic. Moreover, circulation is not necessarily seasonal in those regions.

It is not clear how this would impact the general interpretation of our results. Also, our paper has now been re-focused to explain why post-covid influenza epidemics have generally not been unusually large or severe. Inclusion of tropical countries is not straightforward because surveillance output in those regions generally more uneven, in many cases due to smaller surveillance capacity but also due to complex patterns of circulation where e.g. varying climatic conditions generate disparate epidemic patterns in different regions within individual countries. This variability substantially complicates the estimation of lull durations, and hence we limited our study to temperate regions because the discrete season-to-season nature of epidemics, combined with high-quality surveillance data, would allow for the most accurate quantification of the effects of lulls on epidemic dynamics.

b. Flunet is not representative of what happens in a country. Large countries with a single NIC may not be nationally representative. Also, reporting is sometime interrupted, which could falsely indicate a lull in activity.

All surveillance data is prone to errors and problems. Flunet is not exempt from this. That said, there is no better global dataset. The countries represented in the analyses were included because their surveillance data was inspected and found to be of high quality, with no periods of missing data, and hence it is highly unlikely that we are mistakenly identifying lulls. The observed consistency of epidemic sizes across countries gives further credibility to the data used.

c. Grouping influenza B lineages ignores that these lineages are antigenically distinct.

We agree that this is less than ideal. However, this is how Flunet aggregates data in many countries.

6. The conclusions ignore evidence from other viruses, such as RSV, that disruptions to usual seasonality has led to “rebound” epidemics of greater intensity and hospitalisation burden than usual; and that epidemics of influenza are unlikely to occur during their “usual” seasonal period. Disruptions to usual seasonality could continue for several years (e.g. DOI: 10.1073/pnas.2013182117). That alone presents a problem for the healthcare system which times vaccination programmes and surge capacity for winter epidemics.

As with the responses to previous reviewers, this evidence was not ignored, but RSV is different virus with very different dynamics. We are now in position where the seasonal influenza epidemics have returned and have not been unusually large or severe. Our manuscript now presents evidence as to why this has been the case.

Reviewer #4 (Remarks to the Author):

Review of Garza et al, “Impacts of the COVID-19 pandemic on future seasonal influenza

epidemics” for Nature Communications.

1. What are the noteworthy results?

The authors set out to answer a most pressing question: will seasonal influenza return more deadly than before, due to its absence during the COVID-19 pandemic period. Based on serology data from healthy adults (a population from an Amsterdam HIV study), the authors find convincingly that the Ab levels in this population did not reduce significantly while influenza was gone, in the 2-year pandemic period. They also review past examples of lulls in influenza virus circulation.

They then conclude, based on these findings, that influenza will be no more severe than previously, nothing to worry about.

We thank the reviewer for their time and consideration. Regarding the end of the last comment – “nothing to worry about” – we are actively concerned about the burden of respiratory disease and always think it is worth worrying about. Our study was meant to show that modelling studies and general concerns about the re-emergence of seasonal influenza might have been overblown and to analyse the issue at hand in a data-driven fashion. For example, rather than assume a rate of waning of immunity in such a period of absent circulation, we sought to actually measure this rate. Indeed, in the post-pandemic era, the predictions of unusually large and severe epidemics have generally not been realized.

But while the authors succeed in demonstrating that Ab levels are not significantly lower among healthy adults, I do not agree with the conclusion. This is because the study does not consider children, the driver of influenza epidemics. Nor does it consider Ab waning among elderly, those who are most likely to suffer severe influenza or even die.

Consider the amount of serious illness/deaths is given by this simple relationship: number of infected x probability of serious illness given infection. This has not been given consideration in this paper. Here is why:

Children could well have Ab waning and thus higher attack rates: It is actually likely that children – who are not done building their robust immunity against influenza – will have lower Ab levels/waning, and thus the epidemic size will be far greater (driven by that age groups, school age children).

Elderly could well have Ab waning: Meanwhile, elderly, the age group that fares worst with influenza, may well have Ab waning. This is what is typically seen for vaccinated elderly, Ab wanes over a 6 month period. They are not studied in a group of healthy (younger) adults as in this study.

Taken together, higher attack rates in children spills over in other age groups. Thus, elderly are at a higher risk for getting infected. And when they do, likely have a higher risk of dying due to Ab waning. This means, one could make a good case that a near future influenza epidemic (especially of H3N2 in which the elderly typically far worst of all subtypes) will cause more infections and more serious illness.

The way to address this would be to consider the extreme age groups. How does immunity build up in children – and at what rates does it typically wane? Perhaps serology studies in children. Next, Ab waning among elderly is a critical aspect of making this argument; set an Ab study in a population over 65, or better, over 75 years of age.

We appreciate the reviewer's concern. Our study did include elderly individuals and the waning rates in this group were highly similar to other adults. These data are shown in Fig 4. Unfortunately, we could not include children. However, if the dynamic of lack of build-up in immunity in younger groups and resultant spillover to the elderly were true and would lead to 1) larger and 2) more severe epidemics (higher mortality rates), this would also have applied in pre-pandemic lulls. However, our epidemiological analysis suggests 1) that the impact on epidemic size is substantially outweighed by other factors and 2) that there was no obvious relationship between lull duration and excess mortality in the pre-pandemic era. The crux, of course, is that the scenario where 'a near future influenza epidemic ... will cause more infections and more serious illness' did not pan out – our manuscript in its current form sets out to explain why not. Indeed, one of the key messages of the manuscript in its new form is that these preconceived notions of how influenza virus epidemiology works do not necessarily concord with reality.

2. Will the work be of significance to the field and related fields? How does it compare to the established literature? If the work is not original, please provide relevant references. The question is of course of relevance to us all – so also to people in their field and related fields, like epidemiology, economics and policy. However, the conclusion is not relevant, unless the concerns above are addressed: will there be higher attack rates among children due to Ab waning? Will elderly have a higher risk of severe influenza due to Ab waning?

When we originally submitted our manuscript, we were still very much in the midst of the pandemic and our predictions were speculative. To this end, we appreciate the reviewer's concerns. However, we are now in a period where influenza epidemics now regularly occur and have not been unusually large or severe. Understanding why the rebound of influenza has not been as severe as predicted is important for all of the fields listed above.

3. Does the work support the conclusions and claims, or is additional evidence needed? No. The study does not support not the claim that seasonal influenza after the pandemic period will not be severe. See my rationale above.

We hope that in light of all of the responses above, and our substantially revised manuscript, that the reviewer might now feel differently.

4. Are there any flaws in the data analysis, interpretation and conclusions? - Do these prohibit publication or require revision?

Yes, it is a problem that children and elderly are not specifically studied.

It is also not clear whether a HIV diagnosis in the study population is interfering with the Ab level findings? Would be preferable that the study population is the control population in the Amsterdam HIV study (if there is one).

One of the cohorts included in our study was a cohort consisting of men who have sex with men. All of the members of that cohort included in our study were HIV-negative. The second cohort included in our study was more generally representative of the population of Amsterdam and was also HIV negative.

Finally, the study does not discuss particularly remarkable events of "lulls", in which seasonal influenza in fact was far more severe after a long absence. A good example is the deadly influenza H3N2 epidemic in Madagascar that occurred in 2002, after a long absence of this subtype from the island, see here: (<https://pubmed.ncbi.nlm.nih.gov/12458917/>).

We are not aware of literature that the reviewer is referring to. Also, the report from Madagascar does report an unusually severe outbreak of influenza, but it is not clear to us how it is related to prolonged absence of circulation. Additionally, our results do not imply that influenza seasons will never be severe after long absences, but our results suggest that if this were to occur it would likely not solely be due to the long absence, but due to a combination of factors – one of the key points of our manuscript is that we lack a good understanding of how the different elements of influenza virus epidemiology interplay to determine epidemic size.

5. Is the methodology sound? Does the work meet the expected standards in your field?

Yes the study is well done for the healthy adult study population, and review of past epidemic lulls. But serology data for extreme age groups (children, elderly) are sorely missing.

6. Is there enough detail provided in the methods for the work to be reproduced?

Yes

REVIEWER COMMENTS

Reviewer #1 (Remarks to the Author):

Review of de Jong et al "Multi-season epidemic forecasting and the impacts of lulls in seasonal influenza virus circulation"

The paper investigates how the size of influenza winter-epidemics depends on previous flue epidemics of the same subtype. Specifically the authors study how the size of a winter epidemic depends on the size of the outbreak 1-2 years earlier and in addition discuss the immuno-dynamics of a cohort of Dutch volunteers that have been followed over several seasons. The paper gives new data and especially a new way to analyze the dynamics of influenza winter-epidemics.

The paper addresses an important and timely question in our understanding of seasonal influenza epidemiology. A popular theory claims that influenza immunity wanes substantially within a couple of seasons such that one would see a major outbreak of a given subtype a few years after the last outbreak with that type. In particular the notion of "immune-debt" following the corona lock-downs has received much attention. The paper provides convincing arguments that this is not the case and that there is no particular reason to expect an immune-debt for influenza following the lock-down period.

The modern view on influenza immune-decay and virus drift would suggest that waning and substantial changes in population level immunity occurs at a somewhat longer time-scale than the two years which is studied in the present paper. The paper suggests methods that could test that view. The duration of immunity is supposed to depend on age and it would have been nice if that aspect could have been addressed in the paper.

I have a few minor questions and comments for the authors consideration:

1. While the substance of the paper is fine and sound, I believe that the paper spends a disproportionate amount of space on bashing on a few light weight modeling papers. I recommend that you reduce that part of the paper - it makes your paper appear as a comment on others work rather than a paper with novel insights of its own.

2. The covid-19 pandemics lead to a substantial increase in testing activity -- also for influenza infection -- as the authors point out. However the pandemic also led to a substantial increase in vaccine uptake - where I come from the uptake was doubled compared to pre covid levels. This could provide an alternative explanation for the moderate influenza epidemics in the 2022-23 season???

3. I am a bit concerned with the analysis of flue B as it seems that the authors combine Yamagata and Victoria -- which I believe should be considered as separate "subtypes" for this analysis?.

4. I would suggest a couple of references: (for details see the end of my review)

A. In the discussion of other sources of large variation between the size of epidemics, I think you ought to refer to Smith et al (2004). It seems quite natural to check if the "cluster jumps" described by Smith et al provide a better explanation for the observed synchronized large outbreaks. Koelle et al (2006) may be relevant for that discussion as well.

B. The duration of immunity, how it wanes, and its importance relative for immune drift of the virus have been studied from a serological perspective in Kurcharski et al (2015). Kurcharski et al found that immunity (in particular among the elderly) wanes over 4-5 years -- somewhat slower than your analysis would pick up.

@article{smith2004mapping,
title={Mapping the antigenic and genetic evolution of influenza virus},
author={Smith, Derek J and Lapedes, Alan S and De Jong, Jan C and Bestebroer, Theo M and Rimmelzwaan, Guus F and Osterhaus, Albert DME and Fouchier, Ron AM},
journal={science},

```
volume={305},
number={5682},
pages={371--376},
year={2004},
publisher={American Association for the Advancement of Science}
}
```

```
@article{koelle2006epochal,
title={Epochal evolution shapes the phylodynamics of interpandemic influenza A (H3N2) in humans},
author={Koelle, Katia and Cobey, Sarah and Grenfell, Bryan and Pascual, Mercedes},
journal={Science},
volume={314},
number={5807},
pages={1898--1903},
year={2006},
publisher={American Association for the Advancement of Science}
}
```

B In the discussion of the role of waning immunity vs immunity drift I would suggest that you include

```
@article{kucharski2015estimating,
title={Estimating the life course of influenza A (H3N2) antibody responses from cross-sectional data},
author={Kucharski, Adam J and Lessler, Justin and Read, Jonathan M and Zhu, Huachen and Jiang, Chao Qiang and Guan, Yi and Cummings, Derek AT and Riley, Steven},
journal={PLoS biology},
volume={13},
number={3},
pages={e1002082},
year={2015},
publisher={Public Library of Science San Francisco, CA USA}
}
```

Reviewer #2 (Remarks to the Author):

The authors present an updated analysis of their previous paper, this time focusing on the post-pandemic dynamics of influenza and comparison to dynamics after lulls in seasonal influenza circulation.

The authors have addressed my previous comments.

My main comment on the new manuscript is that despite 'Multi-season epidemic forecasting' being in the title, there is actually relatively little about forecasting in the manuscript, other than some brief discussion points - most of the analysis focuses on effect size estimates (which is useful, but not the same objective). If the authors claim a 'lack of understanding of many aspects of influenza epidemiology fundamentally hampers our ability to generate meaningful forecasts of epidemic size and severity', it would be worth defining what is meant by 'meaningful' in terms of common metrics, perhaps drawing on the extensive literature around forecasting for influenza and other pathogens. In particular, how should the differences in Supplementary Figure 1 be interpreted? How 'wrong' would we be if the models without seasonal effects were used to make multi-season predictions?

In addition, Figure 4 is also mentioned very briefly in the results (on line 305). It would be useful to have some additional motivation/interpretation of this figure if it is to form part of the main results, rather than being a supplementary check. E.g. what is the benefit of knowing there is no age or sex-related effect on baseline titres or waning rates? Similarly, how should be

measurement error be interpreted? 5-11% seems quite low, so presumably the point is that this won't influence conclusions?

Reviewer #3 (Remarks to the Author):

The paper has attempted to demonstrate that a rebound influenza epidemic should not have been expected at the end of the COVID-19 pandemic because there are always lulls in activity of individual influenza viruses. Despite the shift in focus from the original submission I'm still not entirely convinced.

1. Lines 60-69: the drops in influenza activity do not acknowledge the omicron epidemics that changed people's behaviour and probably competed for the same niches as influenza. The text specifically refers to the Australian influenza season, but that season ended concurrent with a rapid rise in sars-cov-2 transmission.
2. Lines 95-104: the lulls described are not the same as the local extinction of influenza that occurred in many parts of the world. It is unclear why a lull would have the same results as a temporary extinction.
3. Line 111 lists 20 countries, but which countries are these? I did not see a list in any of the tables of figures. Further, the Discussion mentions 47 countries. So, which is it? It would be helpful if the supplementary data included the summary tables for each country.
4. Figure 1 d and e: I did not understand the explanation of the points and the plots are quite messy. Consider rephrasing and redrawing.
5. Lines 200-220: mortality is only one measure of the severity of disease and it is closely tied to the viruses that infect elderly adults, not children. It would be more convincing if similar data were available for hospital or ICU admissions.
6. Line 238-241: how many men and how many women were in the RECOVERED cohort? Can this be added to the Fig 3 caption (i.e. instead of "65 RECOVERED participants" state XX men and XX women...). A table summarizing these patients would help.
7. When were samples collected for the RECOVERED study? If they were at random times then vaccination status might not matter, but if they were collected at certain times of the year vaccination status will matter. Is there any reason to believe these people do not represent the source population in terms of vaccination coverage? E.g. if they are covid patients are they unvaccinated for covid and is there covid vaccination status correlated with their flu vaccination status?
8. Fig 3. The caption might have confused parts a and b. Can the antigens used in each year be stated somewhere, preferably in the X axis (using abbreviated names). I don't understand why there was so much more Yamagata in 2018 than H1 and H3 combined. Is the scale correct for this plot? There is a typo: "tires" should be "titres".
9. Fig 4. Provide more details in the caption about the plots shown. Are these geom_smooth and therefore LOESS curves? Are the assumptions for that type of regression appropriate for these data?
10. In all plots and figure showing titre values rather than log2 titres would be easier for interpretation.
11. Line 341: is "manifold" the correct term? Perhaps multifactorial would be more appropriate.
12. Line 375: the ILI and percent positive data are discussed like they were treated separately, but the two were multiplied to get a proxy of influenza activity, correct?
13. Other limitations of ILI data are that sentinel sites are not always constant and have doctors drop in and out.
14. Lines 400-404: I don't agree that we would expect the serological analysis in children to be no different from adults. It is likely to be quite different, as previously discussed. The response to this comment was not adequate.

Reviewer #4 (Remarks to the Author):

This is my second review of this manuscript, so I will be brief. The paper is now more acceptable for publication in Nature Comm. The argument is strengthened by the actual experienced absence of a severe return of seasonal influenza after the covid-19 pandemic.

I am happy to see the caveat about lack of children in the study.

I remain however puzzled about the authors displaying a lack of interest in looking at the severity of historical influenza epidemics in remote areas. The present analysis is mostly concerned iwth European data and an absence of influenza variants of at most 3 years. But some remote settings could yield key information about absence of influenza in more than 3 years. An example is the outbreak Madagascar in 2002, which influenza folks will remember as severe -- so severe in fact that a large outbreak investigation was orchestrated out of concern that this was an unusual influenza strain. I am providing another link to this important event -- in it are references to other historical influenza outbreaks in remote areas.

<https://www.cdc.gov/mmwr/preview/mmwrhtml/mm5145a2.htm>

Indeed, would it not be likely that 3 years absence is too little but more years of absence would provide substantial waning of immunity?

REVIEWER COMMENTS

Reviewer #1 (Remarks to the Author):

Review of de Jong et al "Multi-season epidemic forecasting and the impacts of lulls in seasonal influenza virus circulation"

The paper investigates how the size of influenza winter-epidemics depends on previous flu epidemics of the same subtype. Specifically the authors study how the size of a winter epidemic depends on the size of the outbreak 1-2 years earlier and in addition discuss the immuno-dynamics of a cohort of Dutch volunteers that have been followed over several seasons. The paper gives new data and especially a new way to analyze the dynamics of influenza winter-epidemics.

The paper addresses an important and timely question in our understanding of seasonal influenza epidemiology. A popular theory claims that influenza immunity wanes substantially within a couple of seasons such that one would see a major outbreak of a given subtype a few years after the last outbreak with that type. In particular the notion of "immune-debt" following the corona lock-downs has received much attention. The paper provides convincing arguments that this is not the case and that is no particular reason to expect an immune-debt for influenza following the lock-down period.

We thank the reviewer for their constructive comments.

The modern view on influenza immune-decay and virus drift would suggest that waning and substantial changes in population level immunity occurs at a somewhat longer time-scale than the two years which is studied in the present paper. The paper suggests methods that could test that view. The duration of immunity is supposed to depend on age and it would have been nice if that aspect could have been addressed in the paper.

We now more explicitly contrast the duration of immunity in children and adults:

Lines 423-426: "Immune dynamics in children are known to differ from those in adults⁴¹, with potentially higher waning rates. This could lead to increased susceptibility to infection, and the duration of protection against infection is known to be shorter in children than in adults^{41,44}."

I have a few minor questions and comments for the authors consideration:

1. While the substance of the paper is fine and sound, I believe that the paper spends a disproportionate amount on space on baching on a few light weight modeling papers. I recommend that you reduce that part of the paper - it makes your paper appear as a comment on others work rather than a paper with novel insights of its own.

We have reduced the emphasis on the individual modelling papers throughout the manuscript. However, it is important to discuss those papers specifically because a key finding of our paper is that common assumptions used in modelling influenza dynamics are incorrect and that these assumptions have led to erroneous conclusions in previous studies.

2. The covid-19 pandemics lead to a substantial increase in testing activity -- also for influenza infection -- as the authors point out. However the pandemic also lead to a substantial increase in vaccine uptake - where I come from the uptake was doubled compared to pre covid levels. This could provide an alternative explanation for the moderate influenza epidemics in the 2022-23 season???

We agree that increases in vaccine uptake could affect the size of subsequent epidemics. However, we looked into changes in vaccine uptake and found that in most countries, changes in vaccine uptake were generally small. In some countries, including the USA, vaccine uptake was even lower than the pre-pandemic period. Sources are in the links below. In some countries, as the reviewer suggests, there might have been such increases, but there is no obvious or consistent pattern across countries. It is thus unlikely that changes in vaccine uptake provide an explanation for the moderate influenza epidemic size and severity observed in 2022 and 2023.

<https://www.nuffieldtrust.org.uk/resource/adult-flu-vaccination-coverage>

<https://www.cdc.gov/flu/fluview/coverage-2022estimates.htm>

<https://www.cdc.gov/flu/fluview/dashboard/vaccination-dashboard.html>

3. I am a bit concerned with the analysis of flue B as it seems that the authors combine Yamagata and Victoria -- which I believe should be considered as separate "subtypes" for this analysis?.

We agree that it would be preferable to present analyses of both lineages individually. However, performing a meaningful analysis of each lineage separately is difficult because, in many countries, the epidemiological data that we rely on does not specify which lineage an influenza B detection belonged to. Further, the number of instances in which each lineage individually substantially circulated would be low, rendering it difficult to identify statistically meaningful parameter estimates for individual lineages. Nonetheless, it is highly unlikely that combining the two influenza B lineages has an effect on the conclusions of the study. There is evidence (e.g. <https://doi.org/10.1093/infdis/jix509>) of cross-lineage protective immunity which, while far different from within-lineage protection, lends credence to the notion of combining them.

4. I would suggest a couple of references: (for details see the end of my review)

A. In the discussion of other sources of large variation between the size of epidemics, I think you ought to refer to Smith et al (2004). It seems quite natural to check if the "cluster jumps" described by Smith et al provide a better explanation for the observed synchronized large outbreaks. Koelle et al (2006) may be relevant for that discussion as well.

We agree that the emergence of novel antigenic variants ('cluster jumps') provides an attractive explanation for observed large outbreaks, and we did point this out in our initial submission ('antigenic novelty', line 342-343) and revised submission (lines 364-365). We also agree that the inclusion of the proposed references is reasonable and have done so. However, previous work has shown that there is no consistent effect of antigenic change on epidemic size suggesting that cluster jumps are incomplete explanation. This point is also made in the discussion (lines 393-395).

B. The duration of immunity, how it wanes, and its importance relative for immune drift of the virus have been studied from a serological perspective in Kurchanski et al (2015). Kurchanski et al found that immunity (in particular among the elderly) wanes over 4-5 years -- somewhat slower than your analysis would pick up.

We have added more context about how our estimates of waning rates fit into the context of previous studies into the duration of protection against infection:

"We showed that waning rates following periods of absent circulation were largely in agreement with waning rates previously reported for adults during regular periods of influenza virus circulation⁴⁰. This lower waning rate is also more consistent with individual-level estimates of the duration of protection against infection by circulating strains⁴¹, and is

lower than was assumed in models used to project post-COVID-19 lull epidemic sizes^{12,13,15}.” (lines 380-385).

```
@article{smith2004mapping,  
title={Mapping the antigenic and genetic evolution of influenza virus},  
author={Smith, Derek J and Lapedes, Alan S and De Jong, Jan C and Bestebroer, Theo M  
and Rimmelzwaan, Guus F and Osterhaus, Albert DME and Fouchier, Ron AM},  
journal={science},  
volume={305},  
number={5682},  
pages={371--376},  
year={2004},  
publisher={American Association for the Advancement of Science}  
}
```

```
@article{koelle2006epochal,  
title={Epochal evolution shapes the phylodynamics of interpandemic influenza A (H3N2) in  
humans},  
author={Koelle, Katia and Cobey, Sarah and Grenfell, Bryan and Pascual, Mercedes},  
journal={Science},  
volume={314},  
number={5807},  
pages={1898--1903},  
year={2006},  
publisher={American Association for the Advancement of Science}  
}
```

B In the discussion of the role of waning immunity vs immunity drift I would suggest that you include

```
@article{kucharski2015estimating,  
title={Estimating the life course of influenza A (H3N2) antibody responses from cross-  
sectional data},  
author={Kucharski, Adam J and Lessler, Justin and Read, Jonathan M and Zhu, Huachen  
and Jiang, Chao Qiang and Guan, Yi and Cummings, Derek AT and Riley, Steven},  
journal={PLoS biology},  
volume={13},  
number={3},  
pages={e1002082},  
year={2015},  
publisher={Public Library of Science San Francisco, CA USA}  
}
```

Reviewer #2 (Remarks to the Author):

The authors present an updated analysis of their previous paper, this time focusing on the post-pandemic dynamics of influenza and comparison to dynamics after lulls in seasonal influenza circulation.

The authors have addressed my previous comments.

My main comment on the new manuscript is that despite 'Multi-season epidemic forecasting' being in the title, there is actually relatively little about forecasting in the manuscript, other than some brief discussion points - most of the analysis focuses on effect size estimates

(which is useful, but not the same objective). If the authors claim a 'lack of understanding of many aspects of influenza epidemiology fundamentally hampers our ability to generate meaningful forecasts of epidemic size and severity', it would be worth defining what is meant by 'meaningful' in terms of common metrics, perhaps drawing on the extensive literature around forecasting for influenza and other pathogens. In particular, how should the differences in Supplementary Figure 1 be interpreted? How 'wrong' would we be if the models without seasonal effects were used to make multi-season predictions?

We understand the reviewer's point about the title. Our initial motivation for the title was that our manuscript is closely tied to forecasting: the forecast resurgence of influenza post-pandemic was the direct motivation for our study. Articulated differently, our study focuses on understanding the determinants of epidemic size, and the implications of these limitations of our understanding for forecasting. In light of the reviewer's comment, we have changed the title to: "Determinants of epidemic size and the impacts of lulls in seasonal influenza virus circulation".

We also expanded on the implications of our study for forecasting, and the implications of not accounting for season effects:

"Crucially, our lack of understanding of many of these determinants currently limits our capacity to generate meaningful multi-year forecasts of epidemic sizes. Predictive models that incorporate the uncertainty arising from the unpredictability of season effects will necessarily yield outputs that have wide confidence intervals, limiting their utility for public health purposes. Simultaneously, not incorporating this uncertainty will likely result in substantial prediction error. It is likely that a better understanding of the different immunological, evolutionary, ecological and epidemiological factors that determine epidemic size, beyond waning immunity, is required to perform accurate and precise multi-year prediction of epidemic size." (lines 367-375).

We further expanded on the interpretation of Supplementary Figure 1 in its caption:

"ELPD quantifies the posterior predictive accuracy of a model, while penalizing model complexity. A higher ELPD means better model fit. Hence, the negative differences suggest that, for all model formulations, models that incorporate season effects have substantially superior predictive accuracy compared to those that do not incorporate season effects."

In addition, Figure 4 is also mentioned very briefly in the results (on line 305). It would be useful to have some additional motivation/interpretation of this figure if it is to form part of the main results, rather than being a supplementary check. E.g. what is the benefit of knowing there is no age or sex-related effect on baseline titres or waning rates? Similarly, how should measurement error be interpreted? 5-11% seems quite low, so presumably the point is that this won't influence conclusions?

We decided to include this figure because reviewers made several comments on our original submission regarding the distribution of sex and age of participants in our cohort, through possible sex- and age-related difference in immune dynamics (specifically reviewer 1 remark 1, reviewer 3 remark 4, and reviewer 4 remark 1 in the previous review). As these comments suggest that the lack of such differences could be an interesting insight, we decided to explicitly explore these issues in the main text. We now make this motivation clearer:

"To investigate potential age or sex-specific patterns in antibody dynamics in our two cohorts, we stratified baseline antibody titres by age and sex, for each (sub)type. However, we found no consistent age- or sex-related effects on baseline titres (Fig. 4a, Supplementary

Fig. 6a). Similarly, we investigated if there were age or sex-specific effects on estimated individual-level waning rates, but we found no consistent age or sex-related differences (Fig. 4b, Supplementary Fig. 6b).” (lines 314-319)

To improve the consistency with Figure 3, we have modified Figure 4 such that the same 70 individuals from the ACS cohort that are presented in Fig. 3 are represented in Fig. 4. Analogously to Supplementary Figure 5, we present individual-level baseline titres and waning rates for individuals 1-30 from the ACS in a newly added Supplementary Figure 6.

We have updated the section on measurement error. We now report the standard deviation of the measured titre around the model-estimated titre, as a more informative and representative term, and expanded on its interpretation:

“The estimated standard deviation of the measured titre value around the model-estimated individual-level titre amounted to 0.31 \log_2 units (95% CI 0.29-0.33) for the ACS cohort for the full 2017-2021 dataset and 0.05 (95% CI 0.03-0.08) for the RECoVERED cohort, suggesting that the model used to estimate the waning rates fits the data well.” (lines 319-323)

Reviewer #3 (Remarks to the Author):

The paper has attempted to demonstrate that a rebound influenza epidemic should not have been expected at the end of the COVID-19 pandemic because there are always lulls in activity of individual influenza viruses. Despite the shift in focus from the original submission I'm still not entirely convinced.

We thank the reviewer for their consideration of the manuscript.

That said, the reviewer's synopsis “because there are always lulls in activity of individual influenza viruses” is an unfair caricature of our study and findings.

1. Lines 60-69: the drops in influenza activity do not acknowledge the omicron epidemics that changed people's behaviour and probably competed for the same niches as influenza. The text specifically refers to the Australian influenza season, but that season ended concurrent with a rapid rise in sars-cov-2 transmission.

We cannot exclude the possibility that the co-circulation of the Omicron variant in Australia may have affected 2022 influenza virus epidemic dynamics. However, the ongoing 2023 Australian influenza season is not impacted by such dramatic co-circulation and it appears to be of similar magnitude and severity to 2022. In both the US and the UK, levels of SARS-CoV-2 transmission were low during the 2022/2023 winter influenza season. Hence, co-circulation of SARS-CoV-2 is highly unlikely to be the cause for the general global observation that the seasonal influenza epidemics following the COVID-19 pandemic were not unusually severe.

In our revised manuscript, we have added the United Kingdom as an additional example:

“In the United Kingdom, too, rates of influenza-like illness (ILI) and influenza-attributable mortality fell well within the range observed in the decade preceding the COVID-19 pandemic¹⁶.” (lines 67-69)

UK flu epidemics: <https://www.gov.uk/government/statistics/annual-flu-reports/surveillance-of-influenza-and-other-seasonal-respiratory-viruses-in-the-uk-winter-2022-to-2023>

US covid: <https://www.worldometers.info/coronavirus/country/us/>

UK covid: <https://www.worldometers.info/coronavirus/country/uk/>

2. Lines 95-104: the lulls described are not the same as the local extinction of influenza that occurred in many parts of the world. It is unclear why a lull would have the same results as a temporary extinction.

The reviewer does not expand on why they believe lulls are not the same as the local extinction observed in many countries during the COVID-19 pandemic. In response, we can only express why they *are* similar.

We define lulls as the absent or near-absent circulation of individual (sub)types in individual countries. If the circulation of individual (sub)types was absent in one or more seasons, this is effectively a local extinction of those (sub)types. In those cases, lulls equate to local extinction of a (sub)type, as seen during the COVID-19 pandemic.

If circulation was near-absent, local extinction indeed did not occur. But, the key hypothesis that was widely referenced during the COVID-19 pandemic was that the absence of circulation of influenza viruses, combined with waning immunity, would lead to larger subsequent epidemics. For the mechanisms that are posited to underlie this, it does not really matter if there was no circulation (i.e. local extinction), or very little circulation. Hence, a lull that did see some influenza circulation but at very low levels would have highly similar results to temporary extinction.

Our reasoning for this is explained on lines 87-92:

“Prior to the COVID-19 pandemic, seasonal influenza virus circulation was highly heterogeneous, with individual influenza epidemics in any given country typically being dominated by one or two influenza virus (sub)types. Hence, there were frequent lull periods lasting 1-3 years where other seasonal influenza virus subtypes barely circulated. Due to the lack of immunological cross-reactivity between (sub)types, these lulls are potentially analogous to the scenario observed in the first two years of the COVID-19 pandemic for individual (sub)types.”

3. Line 111 lists 20 countries, but which countries are these? I did not see a list in any of the tables of figures. Further, the Discussion mentions 47 countries. So, which is it? It would be helpful if the supplementary data included the summary tables for each country.

We have added a supplementary table that denotes specifically which countries are included in the analyses (Supplementary Table 1). The discrepancy between the 47 and 20 countries is due to the fact that of the 47 countries used to investigate the frequency of lulls, only 20 were retained for the explicit quantification of epidemic size; these 20 countries were chosen because out of the 47 countries, they had high-quality ILI data for the entire period from 2010 to 2019, and were located in the same hemisphere. We discuss the two datasets extensively in the Methods, and note this point: “We limited this dataset to countries for which influenza-like illness (ILI) records were available for all seasons from 2010-2011 until 2019-2020, and for which virological surveillance data was available, as described above”. (lines 474-476). Nevertheless, we have made clearer this point in the main text:

“To that end, we estimated (sub)type-specific relative epidemic sizes by integrating virological surveillance data with influenza-like illness (ILI) data from the WHO FluID18 database for 20 countries in Europe and the Middle East where, in addition to the virological surveillance data described above, high-resolution ILI data was available (Supplementary Table 1).” (lines 110-114)

4. Figure 1 d and e: I did not understand the explanation of the points and the plots are quite messy. Consider rephrasing and redrawing.

We have modified the plots and relevant captions. We hope that these are now clearer. We have also revised all figures to improve readability and ensure that they comply with *Nature Communications* guidelines.

While doing so, we realized that the waning rates reported in the main text (lines 285-293) did not match those in Figure 3, which showed the correct values. We have now corrected these values in the text. Importantly, the interpretation of the data remains unchanged: antibody waning during and before the COVID-19 pandemic was highly limited. The results are now consistent between all figures, the main text, the methods and the source data. We have modified the text to reflect the changes:

“We applied a mathematical model on the HI titres of participants in 2020 and 2021 to estimate pandemic-period antibody titre waning rates. For the ACS individuals, we estimated that antibody titres against A/H3N2 viruses waned at $-0.06 \log_2$ units per year, 95% credible interval (CI) $(-0.18, 0.05)$; A/H1N1pdm09 viruses at -0.01 , 95% CI $(-0.14, 0.13)$; B/Yamagata viruses at 0.10 , 95% CI $(-0.02, 0.22)$; and B/Victoria viruses at 0.10 , 95% CI $(-0.04, 0.24)$ (Fig. 3d, Supplementary Fig. 5d). For the RECoVERED cohort, we estimated mean waning rates towards A/H3N2, A/H1N1pdm09, B/Yamagata, and B/Victoria to be -0.15 , 95% CI $(-0.32, 0.02)$, -0.06 , 95% CI $(-0.28, 0.17)$, -0.08 , 95% CI $(-0.21, 0.05)$ and -0.11 , 95% CI $(-0.24, 0.02)$ \log_2 units per year respectively, in agreement with those derived from the ACS cohort (Fig. 3d). Combining data from both cohorts for the 2020-2021 period, the estimated mean waning rates remained similar to previous estimates (Fig. 3d). We also estimated mean waning rates using HI titres from the same ACS individuals for the entire 2017-2021 period (Fig. 3d, Supplementary Fig. 5d). For this period, waning estimates are generally lower with narrower credible intervals as they were estimated from longitudinal data spanning five years, but no substantial waning of HI titres against any of the viruses was observed either, and estimates were similar to estimates for the 2020-2021 period, for both the ACS and the RECoVERED cohorts (A/H3N2: -0.20 , 95% CI $(-0.25, -0.15)$, A/H1N1pdm09: -0.09 , 95% CI $(-0.15, -0.04)$, B/Yamagata: -0.13 , 95% CI $(-0.17, -0.08)$, B/Victoria: -0.14 , 95% CI $(-0.19, -0.09)$). For the ACS cohort, we included only individuals who experienced no $\geq 2 \log_2$ unit increases in HI titre for the entire study period and hence were likely not infected in the 2017-2021 period in our waning model (A/H3N2: $n = 59$, A/H1N1pdm09: $n = 54$, B/Yamagata: $n = 53$, B/Victoria: $n = 58$).” (lines 292-313)

5. Lines 200-220: mortality is only one measure of the severity of disease and it is closely tied to the viruses that infect elderly adults, not children. It would be more convincing if similar data were available for hospital or ICU admissions.

Europe-wide hospital or ICU admission data would be nice to have but it is not collated anywhere and very difficult to piece together. We believe it is highly unlikely that hospitalization or ICU data would yield a substantially different picture compared to the mortality data used.

Specifically, we looked into rates of influenza hospitalization in the USA to test the notion that dynamics of severe disease might be distinct in children when looking at the effects of lulls in influenza virus circulation. We found that the rate in individuals aged 0-4 per 100,000 in 2022/2023 was highly similar to rates observed in pre-pandemic seasons (2022/2023: 81, 2019/2020: 91.4, 2018/2019: 70.9, 2017/2018: 71). Of course, this is only for a single country. But, it does suggest that a focus on child-specific rates of severe disease would likely not yield a different conclusion.

(source: <https://gis.cdc.gov/GRASP/Fluview/FluHospRates.html>)

6. Line 238-241: how many men and how many women were in the RECoVERED cohort? Can this be added to the Fig 3 caption (i.e. instead of “65 RECoVERED participants” state XX men and XX women...). A table summarizing these patients would help.

The RECoVERED cohort consists of 34 men and 31 women. We now mention the sex makeup of the RECoVERED cohort in the main text (lines 241-242) and in the caption for Figure 3 (lines 277-278). We have included a supplementary figure that details the sex and age distribution of the participants in both cohorts (Supplementary Fig. 3).

7. When were samples collected for the RECoVERED study? If they were at random times then vaccination status might not matter, but if they were collected at certain times of the year vaccination status will matter. Is there any reason to believe these people do not represent the source population in terms of vaccination coverage? E.g. if they are covid patients are they unvaccinated for covid and is there covid vaccination status correlated with their flu vaccination status?

All RECoVERED subjects were confirmed to not be vaccinated for influenza in 2020. Sample collection took place in the period between June and August 2020, but due to the above has no bearing on the results. We have made this clearer in the text. (line 242)

Selection of the RECoVERED participants for inclusion in our study focused on individuals who were not vaccinated against influenza in the years preceding our study because vaccination could obscure the waning dynamics that the study seeks to address. Beyond vaccination status, the RECoVERED cohort is highly representative of the source population precisely because it was initiated as a population-based cohort that was meant to reflect the local population.

Inclusion for this cohort took place in 2020, when there was no COVID vaccine available, and there is no data to suggest a correlation between COVID vaccination status and influenza vaccination status.

The ACS cohort is more likely to mirror population-level vaccination coverage, as vaccination status was not known. As we note in the manuscript, and for reasons described above, this is a limitation of this cohort.

8. Fig 3. The caption might have confused parts a and b. Can the antigens used in each year be stated somewhere, preferably in the X axis (using abbreviated names). I don't understand why there was so much more Yamagata in 2018 than H1 and H3 combined. Is the scale correct for this plot? There is a typo: “tires” should be “titres”.

We thank the reviewer for correction of the figure captions. We also now state the specific antigens in the figure caption.

It is indeed surprising that the 2017/2018 B/Yamagata season was so large, but this was observed across Europe in the 2017/2018 season (https://www.ecdc.europa.eu/sites/default/files/documents/AER_for_2017-seasonal-influenza.pdf) – this is also precisely why the season effect for influenza B in 2017/2018 is so high (Figure 2).

Following the reviewer's comment, we have modified the y-axis label and caption for Fig. 3a to reflect that these rates are based on the number of *reported* cases of ILI, which depends

on the number of participating sentinel sites. Hence, these do not reflect the true population-level rate of ILI. We also corrected the typo.

9. Fig 4. Provide more details in the caption about the plots show. Are these `geom_smooth` and therefore LOESS curves? Are the assumptions for that type of regression appropriate for these data?

We have provided more details about the plots in the captions:

“**a**, A cross section of antibody titres in 2021 for 70 ACS individuals and 65 RECoVERED individuals, broken down by (sub)type, age and sex. The smoothing line corresponds to a LOESS fit with $\text{span} = 0.75$, for each sex individually. The confidence band corresponds to a 95% confidence interval. **b**, Individual-level fitted waning rates with 50% (thick lines) and 95% (narrow lines) CIs for the 2017-21 period for the individuals among the 70 ACS individuals that did not see a ≥ 4 -fold increase in titre in consecutive years and the 65 RECoVERED individuals for 2020-21, broken down by (sub)type, age and sex. The smoothing line corresponds to a LOESS fit with $\text{span} = 0.75$, for each sex individually. The confidence band corresponds to a 95% confidence interval.” (lines 325-334)

The curves are indeed LOESS curves. A nonparametric method like LOESS is appropriate because from inspecting the raw data it is not clear what the expected relationship between the two variables (age, titre) is likely to be – and indeed there is no clear relationship. There is similarly no clear relationship appear when using linear or polynomial regression instead.

10. In all plots and figure showing titre values rather than \log_2 titres would be easier for interpretation.

Presenting HI titres as \log_2 values has been a standard practice in influenza virus research for ~20 years. It is not clear to us how plotting raw titers would be more clear. Smith et al, *Science*, 2004 (see ref suggestion from reviewer 1) showed that \log_2 titers can be directly projected into a low dimensional Euclidean space with a direct correspondence to antibody cross-reactivity. One of the reasons that paper is so highly cited is precisely because showing HI titres in \log_2 is much easier to interpret than in there raw, un-logged form. Additionally, showing titre values rather than \log_2 titres would cause space issues when the titre value is presented on the x-axis, such as in Fig. 3c, particularly for titre values in the hundreds. For that reason, and to ensure consistency throughout, we use \log_2 titres. Importantly, both titre values and \log_2 titres are used in the literature.

11. Line 341: is “manifold” the correct term? Perhaps multifactorial would be more appropriate.

Manifold, literally meaning ‘many and various’, is the correct term.

12. Line 375: the ILI and percent positive data are discussed like they were treated separately, but the two were multiplied to get a proxy of influenza activity, correct?

Correct, but the percent positive data is first treated separately (for 47 countries). Then, for a subset of 20 of those countries (the ones with high-quality ILI data), the two were multiplied to get the proxy of influenza activity. This is why we discuss the limitations of each data source individually.

13. Other limitations of ILI data are that sentinel sites are not always constant and have doctors drop in and out.

This is a limitation of ILI data, but we carefully chose the 20 countries included because we inspected the time series from each country and only included the countries for which data appeared to have been consistently collected: for example with no missing data and sufficient absolute numbers. It is possible that individual sentinel sites were not constant, but because our analysis relies on the combined data from 20 countries (for which the season sizes are observed to be highly consistent), and because our analyses make use of rigorous statistical methods, errors arising from such biases are accounted for our analyses and are highly unlikely to bias our conclusions. Nevertheless, we have added this limitation of ILI data to the text:

“In particular, the FluID ILI data is not influenza-specific and might be affected by year-to-year variation in sentinel sites, and FluNet data might be biased due to the presence of convenience samples and overrepresentation of outpatient surveillance.” (lines 398-401)

14. Lines 400-404: I don't agree that we would expect the serological analysis in children to be no be different from adults. It is likely to be quite different, as previously discussed. The response to this comment was not adequate.

This comment does not reflect the content of our manuscript. We very clearly stated in the Discussion section of our previous submission: “Immune dynamics in children are known to differ from those in adults, with potentially higher waning rates, which could lead to increased susceptibility to infection.” (lines 397-399).

In the Discussion of the previous and revised version of our manuscript, we argued that the absence of a serological analysis in children, even if it might show that waning rates are higher, is highly unlikely to affect the conclusions of the paper because the dynamics of waning in children are encapsulated in the epidemiological analysis part of the paper. During historical lulls, waning in children also took place and also affected the epidemiological dynamics that we used to derive the effects of lulls on subsequent size. Hence, regardless of the exact waning rates in children, their measurement would not change the conclusions of the study. We have expanded on this paragraph in the Discussion to make this point clearer:

“Due to the lack of children in our serological analysis, the extent to which their waning rates changed during the COVID-19 pandemic remains uncertain. Immune dynamics in children are known to differ from those in adults, with potentially higher waning rates. This could lead to increased susceptibility to infection, and the duration of protection against infection is known to be shorter in children than in adults. Furthermore, the accrual of additional birth cohorts during prolonged periods of absence of influenza virus circulation might affect epidemic dynamics. However, the same dynamics of waning in children and population turnover also occurred in pre-pandemic (sub)type lulls. Although we could not perform the serological analysis for children, the epidemiological data we use to estimate the effects of (sub)type lulls on subsequent epidemic size does incorporate dynamics of waning in children and population turnover. For this reason, the absence of child sera is unlikely to bias our conclusions.” (lines 422-433)

Reviewer #4 (Remarks to the Author):

This is my second review of this manuscript, so I will be brief. The paper is now more acceptable for publication in Nature Comm. The argument is strengthened by the actual experienced absence of a severe return of seasonal influenza after the covid-19 pandemic.

I am happy to see the caveat about lack of children in the study.

We thank the reviewer for their constructive feedback.

I remain however puzzled about the authors displaying a lack of interest in looking at the severity of historical influenza epidemics in remote areas. The present analysis is mostly concerned iwth European data and an absence of influenza variants of at most 3 years. But some remote settings could yield key information about absence of influenza in more than 3 years. An example is the outbreak Madagascar in 2002, which influenza folks will remember as severe -- so severe in fact that a large outbreak investigation was orchestrated out of concern that this was an unusual influenza strain. I am providing another link to this important event -- in it are references to other historical influenza outbreaks in remote areas. <https://www.cdc.gov/mmwr/preview/mmwrhtml/mm5145a2.htm>

Indeed, would it not be likely that 3 years absence is too little but more years of absence would provide substantial waning of immunity?

We thank the reviewer for the added context. We have new added a discussion on dynamics of influenza in remote regions and the possible effects of longer lulls:

“Nevertheless, it is possible that periods of absent circulation much longer than seen during the COVID-19 pandemic will have substantial effects on epidemic dynamics. While the precise mechanisms are unknown and likely multifactorial, the prolonged absence of influenza virus circulation may have contributed to historical observations of severe outbreaks in remote areas, and it is possible that such instances may yield insights into the effects of longer lulls^{29,30}.”(lines 357-362).

REVIEWERS' COMMENTS

Reviewer #4 (Remarks to the Author):

The manuscript is improved, referee concerns have been addressed - I say this manuscript is ready for publication